# Geometric Analysis of Nonlinear Manifold Clustering

**Nimita Shinde**[*]
Lehigh University
nis623@lehigh.edu

**Tianjiao Ding**[*]
University of Pennsylvania
tjding@upenn.edu

**Daniel P. Robinson**
Lehigh University
daniel.p.robinson@lehigh.edu

**René Vidal**
University of Pennsylvania
vidalr@upenn.edu

## Abstract

Manifold clustering is an important problem in motion and video segmentation, natural image clustering, and other applications where high-dimensional data lie on multiple, low-dimensional, nonlinear manifolds. While current state-of-the-art methods on large-scale datasets such as CIFAR provide good empirical performance, they do not have any proof of theoretical correctness. In this work, we propose a method that clusters data belonging to a union of nonlinear manifolds. Furthermore, for a given input data sample $y$ belonging to the $l$th manifold $\mathcal{M}_l$, we provide geometric conditions that guarantee a manifold-preserving representation of $y$ can be recovered from the solution to the proposed model. The geometric conditions require that (i) $\mathcal{M}_l$ is well-sampled in the neighborhood of $y$, with the sampling density given as a function of the curvature, and (ii) $\mathcal{M}_l$ is sufficiently separated from the other manifolds. In addition to providing proof of correctness in this setting, a numerical comparison with state-of-the-art methods on CIFAR datasets shows that our method performs competitively although marginally worse than methods without theoretical guarantees.

## 1 Introduction

Manifold clustering is a fundamental problem in data science, in which one seeks to cluster data lying close to a union of *low-dimensional* manifolds. It has a vast number of applications, such as clustering i) image pixels [1–3], ii) video frames [4, 5], iii) images of faces [6], hand-written digits [7] or other natural objects [8], iv) rigid-body motions [9], v) human actions [10–12], and vi) searching policies for robots [13], to name a few.

Over the past two decades, there has been a considerable amount of work on *subspace clustering*, the special case where each manifold is a *linear or affine subspace*. A key to the success of many of these works can be attributed to the idea of *self-expressiveness* [14]: one can write a data point as linear (or affine) combinations of other points, i.e., given a data point $y$ and matrix of data points $X$, it holds that $y = Xc$ for some vector $c$. The observation that the support of the sparsest such $c$ should correspond to data points in the same subspace as $y$ prompted the study of the optimization problem

$$\min_c \ r(c) + \frac{\lambda}{2}\|e\|_2^2 \quad \text{subject to (s.t.)} \quad e = y - Xc, \tag{SC}$$

where $r(\cdot)$ is a regularizer on $c$ (e.g., $\|\cdot\|_1$ to promote sparsity [6]) and $\lambda > 0$ is a parameter that balances the two terms in the objective. By solving an instance of (SC) for each point in the dataset, one obtains one coefficient vector $c$ per point, and the matrix of all coefficients for all points can be

---

[*]These authors contributed equally to this work

38th Conference on Neural Information Processing Systems (NeurIPS 2024).

used to build a similarity graph of the points and run spectral clustering to obtain a clustering of the data. This has spurred a fruitful line of research, leading to various formulations based on different regularizers [6, 14–20], efficient algorithms [18, 21, 22], and theoretical guarantees on $c$ having the correct support [23–32].

While many interesting datasets (nearly) satisfy the linear or affine subspace assumption, there are a variety of tasks with associated datasets that grossly violate it. For example, natural image datasets such as CIFAR [33] and ImageNet [34] cannot be well modeled by low-dimensional subspaces. Instead, it is more natural to assume that each cluster is modeled by a *smooth low-dimensional non-linear manifold*, a more general case of manifold clustering. Notably, this is much more challenging than subspace clustering, as the global linear relationship among points in each subspace is absent.

Although a few nonlinear manifold clustering methods have achieved high clustering accuracy on large-scale datasets, they lack a theoretical justification. For example, the work in [35–53] aims to learn an embedding (or kernel) via neural networks with subspace clustering style loss functions on the embedded data. While these methods have progressively improved the state-of-the-art in clustering performance, with the most recent ones achieving over $89\%$ accuracy on CIFAR-10 [54–56], little is theoretically understood about why these methods work. In fact, the work of [57] argues the opposite, that some of these methods are provably ill-formulated and learn trivial embeddings [*].

The current paper provides an approach that is both theoretically grounded and empirically tested on modern large-scale datasets. An interesting method that motivates us is *sparse manifold clustering and embedding* (SMCE) [58]. It views a *local neighborhood* of a manifold approximately as a low-dimensional affine subspace, and solves a modified version of (SC) that reweights data and adds an affine constraint $\mathbf{1}^T c = 1$. Despite its effectiveness on simpler datasets such as Extended Yale-B [59], COIL20 [60], and MNIST [7] as observed by [61], so far there is no theoretical understanding of when this method will succeed, nor has it been applied to large-scale datasets.

Nevertheless, the method SMCE inspires our work. A major difficulty in providing theoretical guarantees for SMCE is to deal with the affine constraint. This motivates us to relax the constraint as a penalization, leading to the following model based on the self-expressiveness of the input sample $y$:

$$\min_c \|Wc\|_1 + \frac{\lambda}{2} \cdot \left[ \|e\|_2^2 + \eta \cdot (1 - \mathbf{1}^T c)^2 \right] \quad \text{s.t.} \quad e = y - Xc, \tag{1.1}$$

where $W$ is a diagonal matrix with $j$-th diagonal entry $w_j$ an increasing function of the Euclidean distance between $y$ and data sample $x_j$. The regularizer $\|Wc\|_1$ was adopted from SMCE to promote sparse solutions with support determined by "close" data points. Comparing (1.1) with (SC), we see two differences: (i) (1.1) replaces $r(c)$ with a new regularizer that is a function of $c$ and $W$, and (ii) (1.1) penalizes violation of the constraint $\mathbf{1}^T c = 1$ in the objective by introducing the term $(1 - \mathbf{1}^T c)^2$. Thus, as $\eta$ goes to infinity, the data sample will be represented by an affine combination of input data samples. It is then natural to ask the following question:

**Question 1.** *Can we provide geometric conditions based on the input data samples and structure of the underlying manifolds that ensure that every optimal solution to* (1.1) *is manifold preserving, i.e., the non-zero entries of an optimal solution $c$ correspond to data from the same manifold as $y$?*

## 1.1 Contributions

In this paper, we propose a model that generates an approximately affine representation of an input data sample and provide an answer to Question 1. Our contributions are summarized below.

- *Formulation:* We propose to perform manifold clustering via the convex optimization problem (1.1), which is based on a self-expressiveness property. Notably, the proposed formulation can be efficiently solved using a fast active-set-based method. We then construct an affinity matrix based on the solutions to (1.1), and then use spectral clustering to cluster the data.

- *Theory:* Using our new formulation, we provide an answer to Question 1 by giving theoretical guarantees for any optimal solution of (1.1) to be manifold preserving (see Definition 1). The results depend on the curvature of the manifold at $y$, the relative location of the data samples in the neighborhood of $y$, i.e., the distribution of the sample in the neighborhood, and the separation between the samples from other manifolds and the neighborhood of $y$.

---

[*]Accuracy and running time of methods with theoretical guarantees of correctness have been reported on only datasets with a small number of samples or clusters, or low ambient dimension; see Appendix C for a review.

- *Experiments:* We compare the performance of the proposed method with state-of-the-art alternatives on CIFAR-10,-20 and -100 datasets. The proposed method performs consistently better than subspace clustering methods, and only marginally worse than methods based on deep networks.

## 1.2 Notations

For $l = 1, \ldots, L$, we let $\mathcal{M}_l$ denote a nonlinear manifold in $\mathbb{R}^D$ with intrinsic dimension $d_l \ll D$. We define $\mathcal{M} = \cup_{l=1}^{L} \mathcal{M}_l$ to be the union of manifolds. Let $N$ data samples be generated from $\mathcal{M}$, which we represent as the columns of $X = [x_1, \ldots, x_N] \in \mathbb{R}^{D \times N}$. Additionally, let $y$ be a new sample generated from the union of manifolds $\mathcal{M}$. Without loss of generality, we assume that $y \in \mathcal{M}_1$. Furthermore, for $l = 1, \ldots, L$, let $\mathbf{M}_l$ denote the set of generated input data samples that lie on $\mathcal{M}_l$, and define $\mathbf{M} = \cup_{l=1}^{L} \mathbf{M}_l$ to be the set of all $N$ data samples. For a given subspace $\mathcal{S}$, we let $\mathbb{P}_{\mathcal{S}}(x)$ denote the orthogonal projection of $x$ onto the subspace $\mathcal{S}$.

**Outline.** In Section 2, we introduce our proposed model, define key quantities used in the analysis, and provide theoretical results for the proposed model. In Section 3, we perform experiments on synthetic data aimed to improve our understanding of the theoretical results, followed by a comparison with other existing methods on real data. We conclude the paper in Section 4.

## 2 Proposed Model and Theoretical Analysis

The model (1.1) we study is obtained by penalizing the affine constraint $\mathbf{1}^T c = 1$ in SMCE [58] with penalty parameter $\eta$. The penalty term in the model (1.1) is equivalent to homogenizing the data samples with homogenization constant $\sqrt{\eta}$. Thus, we we propose to study the equivalent problem

$$\min_c \ \|Wc\|_1 + \frac{\lambda}{2}\|\overline{e}\|_2^2 \quad \text{s.t.} \quad \overline{e} = \overline{y} - \overline{X}c \tag{WMC}$$

where $\lambda > 0$, $W$ is a diagonal matrix with positive entries that depend on the Euclidean distances between $y$ and the input data samples, $\overline{X} \in \mathbb{R}^{(D+1) \times N}$ is the matrix whose $j$-th column, $\overline{x}_j$, is the homogenized data sample $(x_j^T, \sqrt{\eta})^T$ with $\eta > 0$ a chosen constant, and $\overline{y} := (y^T \sqrt{\eta})^T$. One could define $w_j := W_{jj} = \|x_j - y\|_2 / (\sum_k \|x_k - y\|_2)$, although other reasonable choices are possible.

### 2.1 Definitions and assumptions

In this section, we state our assumptions and define key quantities used in the theoretical results. Our theoretical analysis provides an answer to Question 1, i.e., we provide conditions under which the non-zero entries of a solution $c^*$ of (WMC) correspond to data samples from the same manifold as $y$. If $c$ satisfies this property, it is called a manifold preserving solution, as we now define.

**Definition 1** (Manifold preserving). *For the data sample $y \in \mathcal{M}_1$, we say that the optimal solution $c^*$ to (WMC) is manifold preserving if and only if $c_j^* = 0$ for all $x_j \notin \mathbf{M}_1$.*

If one considers solving $N$ instances of problem (WMC) (each instance is defined by replacing $y$ with $x_j$, and removing $x_j$ from $X$), then the $N$ solutions may be used to define a similarity graph. If the solution to each instance is manifold preserving, then the similarity graph will only have intra-cluster connections (no inter-cluster/false connections). Consequently, it would be expected that applying spectral clustering to such a graph will result in correct clustering of the data.

We now introduce key quantities and assumptions required to prove the results in Sections 2.2 and 2.3, i.e., to derive conditions under which any optimal solution $c^*$ to (WMC) is manifold preserving.

**Assumption 1.** *We assume that each manifold is smooth and the input data samples generated on each manifold are noiseless. Furthermore, we assume that at least $d_l + 2$ samples are generated on each manifold $\mathcal{M}_l$, where we recall that $d_l$ is the intrinsic dimension of manifold $\mathcal{M}_l$.*

**Definition 2** (Set of nearest neighbors $\mathbf{N}$). *For the data sample $y \in \mathcal{M}_1$, define $\mathcal{B}$ to be the $D$-dimensional ball of smallest radius centered at $y$ such that it contains $d_1 + 1$ points from $\mathbf{M}_1$, where $d_1$ is the dimension of manifold $\mathcal{M}_1$. Define $\mathbf{N}$ to be the set of all data samples in $\mathcal{B}$ excluding $y$, and note that $\mathbf{N}$ may contain samples from both $\mathbf{M}_1$ and $\cup_{l=2}^{L} \mathbf{M}_l$.*

**Definition 3** (Nearest affine subspace $\mathcal{S}$)**.** *For the data sample $y \in \mathcal{M}_1$ and $\mathbf{N}$ in Definition 2, we define the nearest affine subspace, $\mathcal{S}$, as the affine subspace generated by the set $\mathbf{N} \cap \mathbf{M}_1$, i.e.,*

$$\mathcal{S} := \mathit{aff}(\{x_j : x_j \in \mathbf{N} \cap \mathbf{M}_1\}). \tag{2.1}$$

*Furthermore, we define $\overline{\mathcal{S}} := \mathit{span}\{\overline{x}_j : x_j \in \mathbf{N} \cap \mathbf{M}_1\}$ as the linear subspace spanned by the homogenized data samples from the set $\mathbf{N} \cap \mathbf{M}_1$.*

**Assumption 2.** *For the affine subspace $\mathcal{S}$ defined in Definition 3, we assume that $\dim(\mathcal{S}) = d_1$. Equivalently, we assume that $\dim(\overline{\mathcal{S}}) = d_1 + 1$.*

The affine subspace $\mathcal{S}$ is spanned by the $d_1 + 1$ points nearest to $y$ on the manifold $\mathcal{M}_1$ (see Figure 1 for an illustration when $\mathcal{M}_1$ is one-dimensional). We can view $\mathcal{S}$ as an approximation of the tangent space to $\mathcal{M}_1$ at $y$.

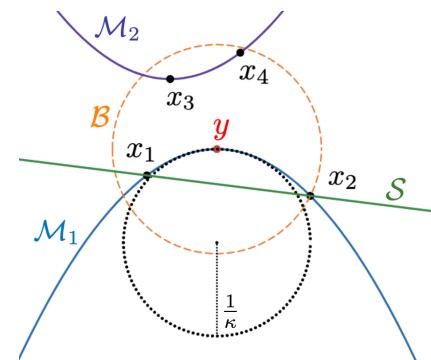

We now define the dual of (WMC) since our analysis requires knowledge of the dual variables.

**Definition 4** (Weighted $\ell_1$ norm and its dual)**.** *The dual norm of the weighted $\ell_1$ norm $\|c\|_{1,W} := \|Wc\|_1$ is*

$$\|c\|_{\infty,W^{-1}} := \max_{\|z\|_{1,W} \le 1} c^T z = \max_{j \in [N]} \frac{|c_j|}{w_j}. \tag{2.2}$$

We can write the dual of (WMC) as

$$\max_{\nu \in \mathbb{R}^{D+1}} \langle \overline{y}, \nu \rangle - \frac{1}{2\lambda}\|\nu\|_2^2 \quad \text{s.t.} \quad \|\overline{X}^T \nu\|_{\infty,W^{-1}} \le 1. \tag{2.3}$$

The constraint in (2.3) can be equivalently written as

$$|\langle \overline{x}_j, \nu \rangle| \le w_j \text{ for all } 1 \le j \le N. \tag{2.4}$$

Figure 1: The affine space $\mathcal{S}$ approximates the tangent to the 1-dimensional manifold $\mathcal{M}_1$ at $y$. Observe that $\{x_i\}_{j=1}^4 \subset \mathbf{N}$ and that $\{x_1, x_2\} \subset \mathcal{M}_1$.

We now define two more quantities (Definitions 5 and (6)) that appear in our main results (see Lemma 1 and Lemma 3). Remark 1 also gives additional insight to these quantities.

**Definition 5** (Dual direction)**.** *For the data sample $y \in \mathcal{M}_1$, define $\overline{X}_{\overline{\mathcal{S}}}$ as the matrix with columns that span the linear subspace $\overline{\mathcal{S}}$ in Definition 3, i.e., $\overline{X}_{\overline{\mathcal{S}}} = [\overline{x}_j]_{x_j \in \mathbf{N} \cap \mathbf{M}_1}$. Define the reduced problem*

$$\min_c \|c\|_{1,W} + \frac{\lambda}{2}\|\overline{y} - \overline{X}_{\overline{\mathcal{S}}} c\|_2^2, \tag{2.5}$$

*and its corresponding dual*

$$\max_\nu \langle \overline{y}, \nu \rangle - \frac{1}{2\lambda}\|\nu\|_2^2 \quad \text{s.t.} \quad \|\overline{X}_{\overline{\mathcal{S}}}^T \nu\|_{\infty,W^{-1}} \le 1. \tag{2.6}$$

*Then, we define the dual direction $\nu^*$ as the unique optimal solution to (2.6).*

The dual direction $\nu^*$ is used to define the separation between the manifold $\mathcal{M}$ and other manifolds. We elaborate on this in Remark 1.

**Definition 6** (Inradius $r(\overline{Q}_{\overline{\mathcal{S}}}^W)$)**.** *For the data sample $y \in \mathcal{M}_1$, define*

$$\overline{Q}_{\overline{\mathcal{S}}}^W = \mathit{conv}\left\{\pm\frac{\overline{x}_j}{w_j} : x_j \in \mathbf{N} \cap \mathbf{M}_1\right\}. \tag{2.7}$$

*We define $r(\overline{Q}_{\overline{\mathcal{S}}}^W)$ as the inradius of the symmetric convex body $\overline{Q}_{\overline{\mathcal{S}}}^W$, i.e., the radius of the largest Euclidean ball contained in $\overline{Q}_{\overline{\mathcal{S}}}^W$. For simplicity, we use the notation, $r^W = r(\overline{Q}_{\overline{\mathcal{S}}}^W)$.*

It follows from Definition 6 that the inradius $r^W$ increases as $\{w_j\}$ decrease. Since $\{w_j\}$ are proportional to the distances between $y$ and the $d_1 + 1$ points closest to $y$ in the set $\mathbf{M}_1$, the inradius increases when more samples are generated from $\mathcal{M}_1$ near $y$ (see Remark 1 for more details).

## 2.2 Manifold-preserving: general geometric conditions

Our first theoretical result provides geometric conditions that ensure that an optimal solution of (WMC), defined for a given input data sample $y$, is manifold preserving (see Definition 1).

**Lemma 1.** *For $y \in \mathcal{M}_1$, input data $X$, and $\lambda > 0$, define the model (WMC) and let Assumptions 1 and 2 hold. Let $\mathcal{S}$ and $\overline{\mathcal{S}}$ be defined as in Definition 3, and $\nu^*$ and $r^W$ be as given in Definitions 5 and 6, respectively. Let $dist(y, \mathcal{S})$ be the Euclidean distance between $y$ and subspace $\mathcal{S}$, and define*

$$\gamma^W = \min_{x_k \in \mathbf{M} \setminus \mathbf{M}_1} \frac{w_k r^W - \frac{|\langle \overline{x}_k, \mathbb{P}_{\overline{\mathcal{S}}}(\nu^*) \rangle|}{\|\mathbb{P}_{\overline{\mathcal{S}}}(\nu^*)\|_2}}{\|\overline{x}_k\|_2}.$$

*The following statements hold.*

*(a) If*

$$\gamma^W > 0 \quad and \tag{2.8}$$

$$dist(y, \mathcal{S}) < \|\overline{y}\|_2 \frac{\gamma^W}{1 + \gamma^W}, \tag{2.9}$$

*then the interval $(\lambda^l, \lambda^u) := \left( \frac{1}{r^W (\|\overline{y}\|_2 - dist(y, \mathcal{S}))}, \frac{\gamma^W}{r^W dist(y, \mathcal{S})} \right)$ is well defined and nonempty.*

*(b) If $\lambda \in (\lambda^l, \lambda^u)$, then every optimal solution to (WMC) is manifold preserving and nonzero.*

To help the reader better under the inequalities in (2.8) and (2.9), we give the following remark.

**Remark 1.** *For the $d_1$-dimensional manifold $\mathcal{M}_1$, conditions (2.8) and (2.9) depend on the following:*

1. ***Inradius $r^W$:*** *$r^W$ (see Definition 6) provides information of the distribution of the $d_1 + 1$ closest data samples to $y$ on the manifold $\mathcal{M}_1$, i.e., the samples belonging to the set $\mathbf{M}_1 \cap \mathbf{N}$. The inequalities in (2.8) and (2.9) are more easily satisfied if $r^W$ increases. From the definition of the inradius $r^W$, it is clear that $r^W$ increases when the samples in the set $\mathbf{M}_1 \cap \mathbf{N}$ are more uniformly distributed. Furthermore, suppose we generate more samples from the manifold $\mathcal{M}_1$ near $y$. Then, the distances of the $d_1 + 1$ samples in the set $\mathbf{M}_1 \cap \mathbf{N}$ from $y$ will decrease, resulting in a larger inradius. Thus, $r^W$ increases when either the number of samples generated from the manifold $\mathcal{M}_1$ near $y$ increases or the samples in the set $\mathbf{M}_1 \cap \mathbf{N}$ are well-distributed.*

2. ***Inner product $|\langle \overline{x}_k, \mathbb{P}_{\overline{\mathcal{S}}}(\nu^*) \rangle|$:*** *For each $x_k \in \mathbf{M} \setminus \mathbf{M}_1$, the inner product defines the separation between $\overline{x}_k$ and the subspace $\overline{\mathcal{S}}$. If the separation between each point $x_k \in \mathbf{M} \setminus \mathbf{M}_1$ and $\overline{\mathcal{S}}$ increases, the inner product decreases. From the definition of $\gamma^W$, it is clear that the inequalities in (2.8) and (2.9) are more easily satisfied when the separation increases.*

Part (a) of Lemma 1 provides conditions (based on the separation of samples belonging to other manifolds from $\overline{\mathcal{S}}$, as well as the distribution of the input data samples in the neighborhood of $y$) for the interval $(\lambda^l, \lambda^u)$ to be nonempty. Note that $\lambda^l$ is a function of $y$ and $X$ while $\lambda^u$ is a function $y, X$, and $\lambda$ since $\gamma^W$ depends on $\nu^*$, which is an optimal solution to (2.6) with parameter $\lambda$.

Part (b) of Lemma 1 states that if the hyperparameter $\lambda$ satisfies $\lambda^l < \lambda < \lambda^u$, then every solution to (WMC) is manifold preserving and nonzero. While we can choose $\lambda > \lambda^l$ based on knowledge of $(y, X)$, to explicitly compute $\lambda$ satisfying $\lambda < \lambda^u$ (when such a value exists) requires defining $\lambda^u$ as a function of $\lambda$. Although Lemma 1 does not provide this relationship, numerical experiments in Section 3.1 on randomly generated data clarify how $\lambda^l$ and $\lambda^u$ change as a function of $(y, X)$ and $\lambda$.

Interestingly, a geometric result in [62] for a *linear* subspace model can be recovered from Lemma 1.

**Corollary 1** (Special case of (WMC))**.** *For $y \in \mathcal{M}_1$, input data $X$, and $\lambda > 0$, define the model (WMC) with $W = I$. Assume that $\|\overline{y}\|_2 = \|\overline{x}_j\|_2 = 1$ for each data sample $x_j$, and that Assumptions 1 and (2) hold. In addition to the quantities defined in Lemma 1, define*

$$\mu = \max_{x_k \in \mathbf{M} \setminus \mathbf{M}_1} \frac{|\langle \overline{x}_k, \mathbb{P}_{\overline{\mathcal{S}}}(\nu^*) \rangle|}{\|\mathbb{P}_{\overline{\mathcal{S}}}(\nu^*)\|_2},$$

*and let $r = r^W \equiv r^I$ since $W = I$. It then follows from Lemma 1 that if*

$$\mu < r \quad and \quad dist(y, \mathcal{S}) < \frac{r - \mu}{1 + r - \mu}, \tag{2.10}$$

then the interval $\left( \frac{1}{r(1-dist(y,\mathcal{S}))}, \frac{r-\mu}{r\,dist(y,\mathcal{S})} \right)$ is well defined and nonempty. Moreover, if $\lambda$ is in this interval, then every optimal solution to (WMC) is manifold preserving and nonzero.

Furthermore, if the manifolds are linear, then $dist(y,\mathcal{S}) = 0$ since $y \in \mathcal{S}$. Thus, in this special case, if $\mu < r$, then for any $\lambda > 1/r$, every optimal solution to (WMC) is subspace preserving and nonzero.

### 2.3 Manifold-preserving: curvature-based geometric conditions

Although we believe the results in the previous section are the first of its kind, they depend on quantities related to projections onto $\overline{\mathcal{S}}$, which we would like to avoid, if possible. To achieve this goal, we require curvature information of the manifold $\mathcal{M}_1$ as well as the reach of the manifold $\mathcal{M}_1$. Let us define the curvature and the reach of a manifold.

**Definition 7** (Curvature of a manifold). *Let $\mathcal{M}_1$ be a smooth manifold with $y \in \mathcal{M}_1$. If $1/\kappa$ is the radius of the largest (internal) tangent circle to $\mathcal{M}_1$ at $y$, then the curvature of $\mathcal{M}_1$ at $y$ is $\kappa$.*

**Definition 8** (Reach of a manifold). *Let $\mathcal{M}_1$ be a smooth manifold. The reach of the manifold $\mathcal{M}_1$, reach($\mathcal{M}_1$), is the largest value such that for any $x \notin \mathcal{M}_1$ and $dist(x, \mathcal{M}_1) \leq reach(\mathcal{M}_1)$, $x$ has a unique projection onto $\mathcal{M}_1$. If $\kappa$ is the curvature of the manifold $\mathcal{M}_1$, then $reach(\mathcal{M}_1) \leq 1/\kappa$.*

In Figure 1, $\kappa$ denotes the curvature of the 1-dimensional manifold $\mathcal{M}_1$ at $y$. The curvature of the manifold at $y$ gives a bound on how fast the direction of a point on the manifold changes with respect to the distance traveled. The larger the change in the direction (or the angle), the larger will be the curvature. For linear manifolds, the curvature is zero.

In Lemma 2, we give a relationship between the curvature of the manifold $\mathcal{M}_1$ at $y$, the radius of the ball $\mathcal{B}$ centered at $y$, and dist($y, \mathcal{S}$). Then, in Lemma 3, we provide geometric conditions based on the curvature that guarantee that any optimal solution of (WMC) is manifold preserving.

**Lemma 2.** *Let Assumptions 1 and 2. For $y \in \mathcal{M}_1$, let $\mathcal{B}$ and $\mathcal{S}$ be as defined in Definitions 2 and 3, respectively. Let $\zeta = \zeta(\mathcal{B})$ be the radius of the sphere $\mathcal{B}$, $\kappa$ be the curvature of the manifold $\mathcal{M}_1$ at $y$, reach($\mathcal{M}_1$) be the reach of the manifold $\mathcal{M}_1$, and dist($y, \mathcal{S}$) be the Euclidean distance between $y$ and the subspace $\mathcal{S}$. If reach($\mathcal{M}_1) \geq \zeta$, then*

$$dist(y, \mathcal{S}) \leq \frac{1}{\kappa} - \sqrt{\frac{1}{\kappa^2} - \zeta^2}. \tag{2.11}$$

Combining Lemma 1 and Lemma 2 gives our final result.

**Lemma 3.** *For $y \in \mathcal{M}_1$, input data $X$, and $\lambda > 0$, define the model (WMC) and let Assumptions 1 and 2 hold. In addition to the quantities defined in Lemma 1, let $\zeta$ be the radius of the sphere $\mathcal{B}$, $\kappa$ be the curvature of the manifold $\mathcal{M}_1$ at $y$, and reach($\mathcal{M}_1$) be the reach of the manifold $\mathcal{M}_1$. The following then hold.*

*(a) If*

$$\gamma^W > 0, \tag{2.12}$$
$$reach(\mathcal{M}_1) \geq \zeta, \quad and \tag{2.13}$$
$$\frac{1}{\kappa} - \sqrt{\frac{1}{\kappa^2} - \zeta^2} < \|\overline{y}\|_2^2 \frac{\gamma^W}{1 + \gamma^W}, \tag{2.14}$$

*then there exists a non-empty interval $(\lambda^l, \lambda^u) := \left( \frac{1}{r^W \left( \|\overline{y}\|_2 - \frac{1}{\kappa} + \sqrt{\frac{1}{\kappa^2} - \zeta^2} \right)}, \frac{\gamma^W}{r^W \left( \frac{1}{\kappa} - \sqrt{\frac{1}{\kappa^2} - \zeta^2} \right)} \right)$.*

*(b) If $\lambda \in (\lambda^l, \lambda^u)$, then every optimal solution to (WMC) is manifold preserving and nonzero.*

The quantity $\gamma^W$ in Lemma 3 captures the notion of distribution of the data samples in the set $\mathbf{M}_1 \cap \mathbf{N}$ as well as separation of the data samples not in $\mathbf{M}_1$ from $\overline{\mathcal{S}}$. The condition (2.12) requires $\gamma^W$ to be positive. Condition (2.13) is related to the sampling density of the manifold $\mathcal{M}_1$ near $y$. If the number of samples generated on $\mathcal{M}_1$ near $y$ increases, then $\zeta$ will decrease. Meanwhile, if the curvature $\kappa$ of the manifold at $y$ is large, $1/\kappa$ will be small, and from Definition 8, we see that reach($\mathcal{M}_1$) will be small. This indicates that for (2.13) to hold, one may need to generate more samples on $\mathcal{M}_1$ near

$y$. Finally, (2.14) provides a lower bound on $\gamma^W$ based on $\kappa$ (the curvature) and $\zeta$ (the number of samples generated on $\mathcal{M}_1$ near $y$). When all three of these conditions hold, we have a non-empty interval $(\lambda^l, \lambda^u)$ as defined in Lemma 3(a). If $\lambda^l < \lambda < \lambda^u$, then Lemma 3(b) ensures that every optimal solution to (WMC) is manifold preserving and nonzero.

## 3    Computational Results

In this section, we first perform experiments on randomly generated data to understand how the quantities in Lemma 1 change as a function of the input data and the hyperparameter $\lambda$ (see Subsection 3.1). Next, in Subsection 3.2, we compare the performance of our model (WMC) with existing methods on CIFAR datasets. Our goal here is to check the performance of (WMC) and to show that while (WMC) does not outperform the state-of-the-art methods, it performs only slightly worse in terms of clustering accuracy.

### 3.1    Experiments on synthetic data and understanding Lemma 1

In this section, we perform two experiments designed to understand how $\lambda^l$ and $\lambda^u$ (see Lemma 1) change as a function of the number of data samples $N$ and hyperparameter $\lambda$ for randomly generated data from two trefoil knots. The noiseless data samples are embedded in $\mathbb{R}^{100}$. The embedding of data involves generating a random orthonormal basis of a subspace in $\mathbb{R}^{100}$, followed by projecting the samples generated from two trefoil knots onto this subspace. We choose $\eta = 1$ and define the matrix $W$ as a diagonal matrix with $j$-th diagonal entry $w_j = \|y - x_j\|_2 / \sum_k \|y - x_k\|_2$.

**Experiment 1:** $\{\lambda^l, \lambda^u\}$ **versus** $\lambda$.    For this experiment, we evaluate how the values of $\lambda^l$ and $\lambda^u$ change when only $\lambda$ changes. We generate $N_1 \in \{120, 150, 160, 200\}$ and $N_2 = 40$ data samples from two non-intersecting trefoil knots of intrinsic dimension $d_1 = d_2 = 1$. We generate 50 different random embeddings of the data samples from the trefoil knots in $\mathbb{R}^{100}$. We plot the mean values and standard deviations of $\lambda^l$ and $\lambda^u$ for different values of $\lambda \in [4.8, 204.8] \times 10^{-4}$ in Figure 2.

Since $\lambda^l$ is independent of $\lambda$, its value is constant for all values of $(N_1, N_2)$. It also appears that $\lambda^u$ is a monotonically increasing function of $\lambda$ that eventually reaches a "steady-state" value. When $N_1 = 120$, we observe that the conditions in Lemma 1(a) are not satisfied for any value of $\lambda$, and that the interval $(\lambda^l, \lambda^u)$ is empty. When $N_2 = 150$, we observe that $\lambda^u > \lambda^l$ for $\lambda > 0.0012$, but for these $\lambda$ values it also holds that $\lambda > \lambda^u$, meaning that Lemma 1(b) is violated for all $\lambda > 0$. When $N_1 \in \{160, 200\}$, we observe that for some values of $\lambda > 0$, there exist non-empty intervals $(\lambda^l, \lambda^u)$ such that $\lambda \in (\lambda^l, \lambda^u)$. Thus, for every such $\lambda \in (\lambda^l, \lambda^u)$, it follows from Lemma 1 that every optimal solution to (WMC) is manifold preserving and nonzero. We can conclude from these experiments that for certain input data $(y, X)$, there exists a range of *acceptable* values of $\lambda$ for which the conditions defined in both parts (a) and (b) of Lemma 1 are satisfied, so that every optimal solution to (WMC) is nonzero and manifold preserving.

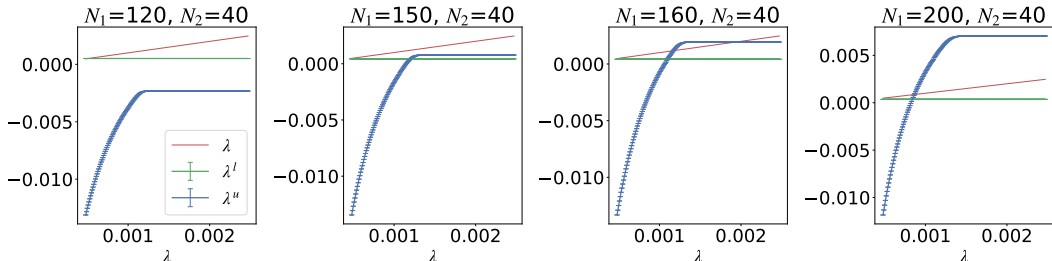

Figure 2: Plot of $\lambda^l$, $\lambda^u$, and $\lambda$ for $(N_1, N_2)$ samples generated from two trefoil knots. The data generated from the knots are embedded in $\mathbb{R}^{100}$ with 50 randomly generated embeddings.

**Experiment 2:** $\{\lambda^l, \lambda^u\}$ **versus** $N$.    In this experiment, we aim to understand how $\lambda^l$ and $\lambda^u$ change when the number of data samples, $N$, changes. We let $N_1 = N_2$ and gradually increase $N = N_1 + N_2$. We generate 50 different random embeddings of data samples from trefoil knots

in $\mathbb{R}^{100}$. For each pair of data $(y, X)$, we compute $\lambda^l$ and then create three values for $\lambda$ by setting $\lambda = \alpha\lambda^l$ for $\alpha \in \{2, 5, 50\}$. Then, for each $\lambda$ value, we plot $\lambda^l$ and $\lambda^u$ versus $N = N_1 + N_2$ in Figure 3. We can observe that as the number of samples increases, the size of the interval $(\lambda^l, \lambda^u)$ increases linearly.

We further observe that for $\alpha = 2$, the interval $(\lambda^l, \lambda^u)$ is non-empty (satisfying the conditions in Lemma 1(a)) for all $N \geq 890$, whereas for $\alpha = 5, 50$, the interval is non-empty for all $N \geq 300$. Thus, we see that the small value of $\alpha$ leads to a slower increase in $\lambda^u$ resulting in the interval $(\lambda^l, \lambda^u)$ becoming non-empty at a much larger value of $N$. Furthermore, we also see that, (i) when $\alpha = 5$, $\lambda \in (\lambda^l, \lambda^u)$ for all $N \geq 330$, and (ii) when $\alpha = 50$, $\lambda \in (\lambda^l, \lambda^u)$ for all $N \geq 510$. In other words, when $\alpha = 5$, the conditions in both parts (a) and (b) of Lemma 1 are satisfied for all $N \geq 330$, whereas when $\alpha = 50$, the conditions in parts (a) and (b) of Lemma 1 are satisfied for all $N \geq 510$, making $\lambda = 5\lambda^l$, the best choice of the hyperparameter in this experiment.

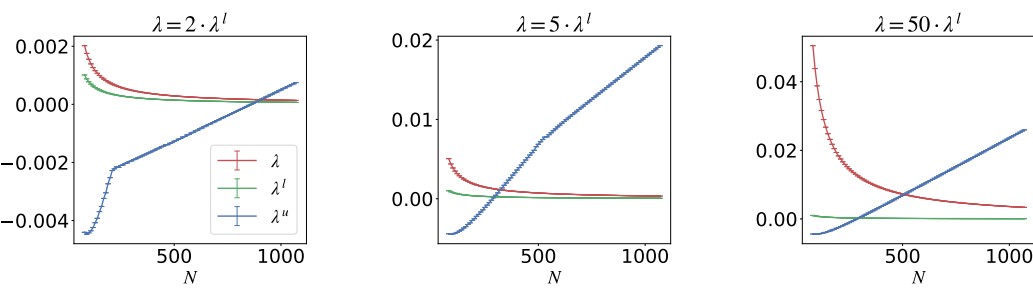

Figure 3: Plot of $\lambda^l$, $\lambda^u$, and $N = N_1 + N_2$ with $N_1 = N_2$ and $\lambda = \alpha\lambda^l$ for $\alpha \in \{2, 5, 50\}$. Samples are generated from two trefoil knots and embedded in $\mathbb{R}^{100}$ with 50 randomly generated embeddings.

## 3.2  Experiments on real data

In this section, we compare the empirical performance of our model with existing methods on CIFAR-10, CIFAR-20, CIFAR-100 datasets [33]. The CIFAR dataset consists of 60000 color images of size $32\times32$ that are divided into 10, 20, and 100 classes for CIFAR-10, CIFAR-20, CIFAR-100, respectively. In Section 3.2.1, we describe the steps for clustering data using our model (WMC), followed by a comparison of empirical results in Section 3.2.2. (See Appendix B.1 for more details.)

### 3.2.1  Clustering data using (WMC)

We follow the steps in Algorithm 1 to cluster the input images using our model (WMC).

---
**Algorithm 1** Pseudocode for clustering data using (WMC)

---
**Input**: Images from CIFAR dataset, $\lambda > 0$, and the number of clusters $L$
 1: Generate features using the CLIP encoder [63].
 2: Construct a representation matrix $C$ by solving problem (WMC) for the CLIP feature vector for each input image.
 3: Construct a symmetric affinity matrix $A$ from $C$.
 4: Apply spectral clustering to $A$ to get the $L$ clusters.
**Output**: $L$ clusters of data

---

The CLIP (Contrastive Language-Image Pre-Training) [63] encoder is a powerful tool in pre-training data. Given its prior success, we use CLIP to map each input image in CIFAR to a feature vector in Step 1 of Algorithm 1. Step 2 of Algorithm 1 employs our model WMC to construct a representation matrix $C$. We now briefly explain this step. For the feature vector $y$ of each input image: (a) we define the positive diagonal distance matrix $W$ so that each diagonal entry is an increasing function of the Euclidean distance between $y$ and each of the other feature vectors, (b) we define the model (WMC) for $\lambda > 0$, (c) we solve (WMC) using an active-set method [18] to (hopefully) obtain a manifold preserving representation for the feature vector of each input image. (The choice of $\lambda$ and $W$ is explained further in Subsection 3.2.2.) Similar to classical spectral clustering methods, the representation matrix is used to define a symmetric affinity matrix $A$; here, we choose $A =$

Table 1: Comparing clustering accuracy (ACC) and Normalized Mutual Information (NMI) for our models L-WMC and E-WMC with state-of-the-art subspace clustering and deep clustering methods. Each method is used to cluster the data pre-trained using CLIP [63]. For each metric (ACC and NMI) and data set, the best overall value achieved is **bolded**, and the best value achieved by our method is in blue.

| Dataset | CIFAR-10 | | CIFAR-20 | | CIFAR-100 | |
| Metrics | ACC | NMI | ACC | NMI | ACC | NMI |
| --- | --- | --- | --- | --- | --- | --- |
| Methods that do not fine-tune representations | | | | | | |
| L-WMC | 96.07 | 90.54 | 63.97 | 69.55 | 69.2 | 76.49 |
| E-WMC | 94.34 | 88.79 | 64.1 | 69.58 | 69.83 | 76.63 |
| SMCE | 86.86 | 90.66 | 63.1 | 70.53 | 68.56 | 77.17 |
| EnSC [18]* | 85.8 | 89.2 | 61.6 | 69.3 | 66.6 | 77.1 |
| SSC [22]* | 85.4 | 84.6 | 60.9 | 65.3 | 64.6 | 72.8 |
| Methods that fine-tune representations | | | | | | |
| TEMI [64]*† | 96.9 | 92.6 | 61.8 | 64.5 | 73.7 | 79.9 |
| CPP [56]*† | **97.4** | **93.6** | **64.2** | **72.5** | **74.0** | **81.8** |

*The numerical results in this row are taken from [56].

† See appendix in [56, 64] for details on fine-tuning models.

$\frac{1}{2}(|C| + |C|^T)$. (We used this definition in combination with other techniques to construct $A$, which are explained in detail in Appendix B.2.) Finally, we use spectral clustering on $A$ to cluster the data.

**Output metrics.** We use two metrics to compare the performance of various clustering methods. In particular, we use Clustering Accuracy and Normalized Mutual Information whose values range from 0 to 100%, with higher values indicating better performance.

### 3.2.2 Numerical Results

In this section, we compare the numerical performance of two instances of our Algorithm 1 with the subspace clustering methods (a) Elastic Net Subspace Clustering (EnSC) [18] and (b) Sparse Subspace Clustering, manifold clustering method (c) SMCE [58], and with the deep clustering methods (d) CPP [56] and (e) TEMI [64]. Each method is applied to the CLIP features extracted from the input images. The two instances of our Algorithm 1 are determined by how the weight matrix $W$ is chosen in (WMC), as we now describe.

- **L-WMC**: *Linearly* Weighted Manifold Clustering model with $w_j = \|x_j - y\|_2 / \sum_k \|x_k - y\|_2$ for all $j$ in (WMC). Here, the weights $\{w_j\}$ are linearly proportional to the Euclidean distances.

- **E-WMC**: *Exponentially* Weighted Manifold Clustering model with weights in (WMC) defined as $w_j = \exp(2\|x_j - y\|_2) / \sum_k \exp(2\|x_k - y\|_2)$ for all $j$. Here, the weights $\{w_j\}$ are defined as an exponential function of the Euclidean distances.

We report the clustering results for L-WMC and E-WMC in Table 1. (See Appendix B.2 for details on the choice of $\lambda$ and $\eta$.) From Table 1, we can observe that the clustering accuracy of L-WMC and E-WMC is consistently better than EnSC, SSC and SMCE for all three datasets. Moreover, L-WMC and E-WMC perform slightly worse than the state-of-the-art method CPP on the CIFAR-10 and CIFAR-20 datasets. So, although our method does not outperform the existing state-of-the-art method, our method is the first to perform similarly to the state-of-the-art methods and have a theoretical guarantee of correctness (see Lemma 1 and Lemma 3).

## 4 Discussion

In this paper, we propose a model whose solution defines a self-expressive representation of the input data, and provide a theoretical analysis of correctness of the model (see Lemma 1 and Lemma 3). For data $(y, X)$ and hyperparameter $\lambda > 0$, we give conditions under which the solution to our model is

non-trivial and manifold preserving. To the best of our knowledge, this is the first work that provides a theoretical understanding of manifold clustering models. We also show that our model performs only marginally worse than the current state-of-the-art methods on CIFAR datasets.

For a given data set, our analysis does not provide a strategy for choosing the hyperparameter $\lambda > 0$ so that it satisfies the conditions in Lemma 1. This leads to an important open question: *For any given data set, can one provide a proof of existence of $\lambda > 0$ for which the conditions in Lemma 1 are satisfied? If so, can one provide a closed form expression to choose the value of $\lambda$?* An answer to this question will result in an "optimal" choice for the value of $\lambda$, thus avoiding the cost of tuning $\lambda$.

## Acknowledgements

We would like to thank the generous support provided by the National Science Foundation's through grants 1704458, 2031985, and 2212457, and the Lehigh internal grant funding the project "Foundations and Applications of Mathematical Optimization and Data Science".

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

# A  Proofs

## A.1  Proof of Lemma 1

*Proof of Lemma 1.* To prove Lemma 1, we start by defining the condition for the solution to (WMC) to be manifold preserving in the following lemma.

**Proposition 1.** *Let $\nu^*$ be the dual direction as defined in Definition 5. If*

$$|\langle \overline{x}_j, \nu^* \rangle| < w_j \quad \forall x_j \in \mathbf{M} \setminus \mathbf{M}_1, \tag{A.1}$$

*then every optimal solution to* (WMC) *is manifold preserving.*

The proof of Proposition 1 is given in Section A.2. Now, from (A.1), we have that

$$|\langle \overline{x}_j, \nu^* \rangle| = |\langle \overline{x}_j, \mathbb{P}_{\overline{\mathcal{S}}}(\nu^*) + \mathbb{P}_{\overline{\mathcal{S}}^\perp}(\nu^*) \rangle| \le |\langle \overline{x}_j, \mathbb{P}_{\overline{\mathcal{S}}}(\nu^*) \rangle| + |\langle \overline{x}_j, \mathbb{P}_{\overline{\mathcal{S}}^\perp}(\nu^*) \rangle|, \tag{A.2}$$

where $\mathbb{P}_{\overline{\mathcal{S}}}(\cdot)$ and $\mathbb{P}_{\overline{\mathcal{S}}_\perp}(\cdot)$ denote the projections on to $\mathcal{S}$ and the subspace orthogonal to $\mathcal{S}$, respectively. Thus, we see that if the stronger condition

$$|\langle \overline{x}_j, \mathbb{P}_{\overline{\mathcal{S}}}(\nu^*) \rangle| + |\langle \overline{x}_j, \mathbb{P}_{\overline{\mathcal{S}}^\perp}(\nu^*) \rangle| < w_j \tag{A.3}$$

is satisfied for all $x_j \in \mathbf{M} \setminus \mathbf{M}_1$, then (A.1) is satisfied.

**Bounding $|\langle \overline{x}_j, \mathbb{P}_{\overline{\mathcal{S}}^\perp}(\nu^*) \rangle|$.**  From KKT conditions, we have that $\nu^* = \lambda(\overline{y} - \overline{X}_{\overline{\mathcal{S}}} c^*)$, where $(c^*, \nu^*)$ is the primal-dual optimal solution to (2.5)-(2.6). So,

$$
\begin{aligned}
\|\mathbb{P}_{\overline{\mathcal{S}}^\perp}(\nu^*)\|_2 &= \|\mathbb{P}_{\overline{\mathcal{S}}^\perp}(\lambda(\overline{y} - \overline{X}_{\overline{\mathcal{S}}} c^*))\|_2 \\
&\underset{(i)}{=} \|\mathbb{P}_{\overline{\mathcal{S}}^\perp}(\lambda \overline{y})\|_2 \\
&= \lambda \mathrm{dist}(\overline{y}, \overline{\mathcal{S}}),
\end{aligned} \tag{A.4}
$$

where (i) follows since $\overline{X}_{\overline{\mathcal{S}}} c^*$ is the linear combination of data samples in $\overline{\mathcal{S}}$ and so, $\overline{X}_{\overline{\mathcal{S}}} c^* \in \overline{\mathcal{S}}$. Furthermore, we have

$$
\begin{aligned}
\mathrm{dist}(\overline{y}, \overline{\mathcal{S}}) &= \min_a \{\|\overline{y} - \overline{z}\|_2 : \overline{z} = \sum_{x_k \in \mathbf{N} \cap \mathbf{M}_1} a_k \overline{x}_k\} \\
&\le \min_a \{\|\overline{y} - \overline{z}\|_2 : \overline{z} = \sum_{x_k \in \mathbf{N} \cap \mathbf{M}_1} a_k \overline{x}_k, \sum_{x_k \in \mathbf{N} \cap \mathbf{M}_1} a_k = 1\} \\
&= \min_a \left\{ \left\| \begin{bmatrix} y \\ \sqrt{\eta} \end{bmatrix} - \begin{bmatrix} z \\ \sqrt{\eta} \end{bmatrix} \right\|_2 : \begin{bmatrix} z \\ \sqrt{\eta} \end{bmatrix} = \sum_{x_k \in \mathbf{N} \cap \mathbf{M}_1} a_k \begin{bmatrix} x_k \\ \sqrt{\eta} \end{bmatrix} \right\} \\
&= \min_a \{\|y - z\|_2 : z = \sum_{x_k \in \mathbf{N} \cap \mathbf{M}_1} a_k x_k, \sum_{x_k \in \mathbf{N} \cap \mathbf{M}_1} a_k = 1\} \\
&= \mathrm{dist}(y, \mathcal{S}).
\end{aligned} \tag{A.5}
$$

From (A.4) and (A.5), we have

$$|\langle \overline{x}_j, \mathbb{P}_{\overline{\mathcal{S}}^\perp}(\nu^*) \rangle| \le \|\overline{x}_j\|_2 \|\mathbb{P}_{\overline{\mathcal{S}}^\perp}(\nu^*)\|_2 \le \lambda \|\overline{x}_j\|_2 \mathrm{dist}(y, \mathcal{S}) \quad \forall x_j \in \mathbf{M} \setminus \mathbf{M}_1. \tag{A.6}$$

Combining (A.3) and (A.6), we have a stronger condition

$$|\langle \overline{x}_j, \mathbb{P}_{\overline{\mathcal{S}}}(\nu^*) \rangle| + \lambda \|\overline{x}_j\|_2 \mathrm{dist}(y, \mathcal{S}) < w_j \quad \forall x_j \in \mathbf{M} \setminus \mathbf{M}_1, \tag{A.7}$$

and equivalently, if

$$\lambda < \frac{w_j - |\langle \overline{x}_j, \mathbb{P}_{\overline{\mathcal{S}}}(\nu^*) \rangle|}{\mathrm{dist}(y, \mathcal{S}) \|\overline{x}_j\|_2} \quad \forall x_j \in \mathbf{M} \setminus \mathbf{M}_1, \tag{A.8}$$

then (A.1) is also satisfied, and every optimal solution to (WMC) is manifold preserving. We can write (A.8) as

$$\lambda < \min_{x_j \in \mathbf{M} \setminus \mathbf{M}_1} \frac{w_j - \frac{|\langle \overline{x}_j, \mathbb{P}_{\overline{\mathcal{S}}}(\nu^*) \rangle|}{\|\mathbb{P}_{\overline{\mathcal{S}}}(\nu^*)\|}}{\mathrm{dist}(y, \mathcal{S}) \|\overline{x}_j\|_2}. \tag{A.9}$$

**Bounding** $\|\mathbb{P}_{\overline{S}}(\nu^*)\|_2$. The dual constraint of (2.6) can be written as

$$
\begin{aligned}
&\|\overline{X}_{\overline{S}}^T \nu^*\|_{\infty, W^{-1}} \leq 1\\
\Rightarrow\ &\|\overline{X}_{\overline{S}}^T(\mathbb{P}_{\overline{S}}(\nu^*) + \mathbb{P}_{\overline{S}^\perp}(\nu^*))\|_{\infty, W^{-1}} \leq 1\\
\Rightarrow\ &\|\overline{X}_{\overline{S}}^T \mathbb{P}_{\overline{S}}(\nu^*)\|_{\infty, W^{-1}} \leq 1\\
\Rightarrow\ &\left|\left\langle \frac{\overline{x}_k}{w_k}, \mathbb{P}_{\overline{S}}(\nu^*) \right\rangle\right| \leq 1 \quad \forall x_k \in \mathbf{M}_1 \cap \mathbf{N}.
\end{aligned}
\tag{A.10}
$$

Recall the definition of the symmetric convex body $\overline{Q}_{\overline{S}}^W = \mathrm{conv}\left\{\pm \frac{\overline{x}_j}{w_j} : x_j \in \mathbf{M}_1 \cap \mathbf{N}\right\}$ as given in Definition 6. From the last inequality in (A.10), we have that $\mathbb{P}_{\overline{S}}(\nu^*)$ belongs to the polar set of $\overline{Q}_{\overline{S}}^W$, i.e., $\mathbb{P}_{\overline{S}}(\nu^*) \in (\overline{Q}_{\overline{S}}^W)^\circ$.

**Proposition 2** ([65]). *For a symmetric convex body $\mathcal{P}$, the inradius of $\mathcal{P}$ and the circumradius of its polar set $\mathcal{P}^\circ$ satisfy*

$$
r(\mathcal{P})R(\mathcal{P}^\circ) = 1.
\tag{A.11}
$$

Using the fact that $\|\mathbb{P}_{\overline{S}}(\nu^*)\|_2 \leq R(\mathcal{K}^\circ)$, and setting $\mathcal{P} = \overline{Q}_{\overline{S}}^W$ in (A.11), we have that

$$
\|\mathbb{P}_{\overline{S}}(\nu^*)\|_2 r(\overline{Q}_{\overline{S}}^W) \leq 1,
\tag{A.12}
$$

or equivalently,

$$
\|\mathbb{P}_{\overline{S}}(\nu^*)\|_2 \leq \frac{1}{r^W}.
\tag{A.13}
$$

Thus, from A.1, (A.9) and (A.13), we see that if

$$
\lambda < \frac{w_j r^W - \frac{|\langle \overline{x}_j, \mathbb{P}_{\overline{S}}(\nu^*)\rangle|}{\|\mathbb{P}_{\overline{S}}(\nu^*)\|_2}}{r^W \mathrm{dist}(y, \mathcal{S})\|\overline{x}_j\|_2} \quad \forall\, x_j \in \mathbf{M} \setminus \mathbf{M}_1,
\tag{A.14}
$$

then every optimal solution to (WMC) is manifold preserving.

However, if $\lambda$ is too small, then the optimal solution to (WMC) will be trivial, i.e., $c = 0$. Thus, to get a non-trivial optimal solution to (WMC), $\lambda$ cannot be too small. The following proposition provides a lower bound on $\lambda$ to get a non-trivial solution.

**Proposition 3.** *If $\lambda > \frac{1}{r^W(\|\overline{y}\|_2 - dist(y,\mathcal{S}))}$, then every optimal solution to (WMC) is nonzero.*

*Proof.* We prove the result by contradiction. Assume that $\lambda > \frac{1}{r^W(\|\overline{y}\|_2 - \mathrm{dist}(y,\mathcal{S}))}$, and $c = 0$ is an optimal solution to (WMC). Then, from KKT condition applied to (WMC), we have that $\nu = \lambda(\overline{y} - \overline{X}c) = \lambda\overline{y}$, where $\nu$ is the optimal solution to the dual (2.3). We have that

$$
\|\overline{X}^T \nu\|_{\infty, W^{-1}} = \lambda\|\overline{X}^T \overline{y}\|_{\infty, W^{-1}} \geq \lambda\|\overline{X}_{\overline{S}}^T \overline{y}\|_{\infty, W^{-1}}.
\tag{A.15}
$$

Furthermore, we have

$$
\begin{aligned}
\|\overline{X}_{\overline{S}}^T \overline{y}\|_{\infty, W^{-1}} &= \|\overline{X}_{\overline{S}}^T(\mathbb{P}_{\overline{S}}(\overline{y}) + \mathbb{P}_{\overline{S}^\perp}(\overline{y}))\|_{\infty, W^{-1}}\\
&= \|\overline{X}_{\overline{S}}^T(\mathbb{P}_{\overline{S}}(\overline{y}))\|_{\infty, W^{-1}}.
\end{aligned}
\tag{A.16}
$$

Combining (A.15) and (A.16), we have

$$
\begin{aligned}
\|\overline{X}^T \nu\|_{\infty, W^{-1}} &\geq \lambda\|\overline{X}_{\overline{S}}^T \mathbb{P}_{\overline{S}}(\overline{y})\|_{\infty, W^{-1}}\\
&= \frac{\lambda\|\overline{X}_{\overline{S}}^T \mathbb{P}_{\overline{S}}(\overline{y})\|_{\infty, W^{-1}}}{\|\mathbb{P}_{\overline{S}}(\overline{y})\|_2}\|\mathbb{P}_{\overline{S}}(\overline{y})\|_2\\
&\geq \lambda \min_{u \in \overline{S}, u \neq 0} \frac{\lambda\|\overline{X}_{\overline{S}}^T u\|_{\infty, W^{-1}}}{\|u\|_2}\|\mathbb{P}_{\overline{S}}(\overline{y})\|_2\\
&\underset{(i)}{=} \lambda r^W \|\mathbb{P}_{\overline{S}}(\overline{y})\|_2,
\end{aligned}
\tag{A.17}
$$

where the last equality follows since the inradius, $r^W$ (as defined in Definition 6), can be written as

$$r^W = \min_{u \in \overline{\mathcal{S}}, u \neq 0} \max_{x_j \in \mathbf{M}_1 \cap \mathbf{N}} \frac{\lambda |\langle \overline{x}_j/w_j, u \rangle|}{\|u\|_2} = \min_{u \in \overline{\mathcal{S}}, u \neq 0} \frac{\lambda \|\overline{X}_{\overline{\mathcal{S}}}^T u\|_{\infty, W^{-1}}}{\|u\|_2}.$$

Furthermore, we have

$$\|\mathbb{P}_{\overline{\mathcal{S}}}(\overline{y})\|_2 = \|\overline{y} - \mathbb{P}_{\overline{\mathcal{S}}^\perp}(\overline{y})\|_2 \geq \|\overline{y}\|_2 - \|\mathbb{P}_{\overline{\mathcal{S}}^\perp}(\overline{y})\|_2 = \|\overline{y}\|_2 - \text{dist}(\overline{y}, \overline{\mathcal{S}}) \geq \|\overline{y}\|_2 - \text{dist}(y, \mathcal{S}), \quad \text{(A.18)}$$

where the last inequality follows from (A.5). Substituting this in (A.17), we have

$$\|\overline{X}_{\overline{\mathcal{S}}}^T \nu\|_{\infty, W^{-1}} \geq \lambda r^W (\|\overline{y}\|_2 - \text{dist}(y, \mathcal{S})) > 1, \quad \text{(A.19)}$$

where the last inequality follows since $\lambda > \frac{1}{r^W(\|\overline{y}\|_2 - \text{dist}(y, \mathcal{S}))}$. Thus, we have that when $\lambda > \frac{1}{r^W(\|\overline{y}\|_2 - \text{dist}(y, \mathcal{S}))}$, $c = 0$ cannot be an optimal solution since the constraint in the dual problem (2.3) is violated. This proves the statement of the proposition. $\qquad \square$

From Proposition 3 and (A.14), we see that if

$$\frac{1}{r^W(\|\overline{y}\|_2 - \text{dist}(y, \mathcal{S}))} < \min_{x_j \in \mathbf{M} \setminus \mathbf{M}_1} \frac{w_j r^W - \frac{|\langle \overline{x}_j, \mathbb{P}_{\overline{\mathcal{S}}}(\nu^*) \rangle|}{\|\mathbb{P}_{\overline{\mathcal{S}}}(\nu^*)\|_2}}{r^W \text{dist}(y, \mathcal{S}) \|\overline{x}_j\|_2}, \quad \text{(A.20)}$$

then we get an interval for acceptable values of $\lambda$ to get a non-trivial and manifold preserving solution to (WMC). Rearranging the terms in (A.20), we see that, if

$$\frac{r^W \text{dist}(y, \mathcal{S})}{r^W(\|\overline{y}\|_2 - \text{dist}(y, \mathcal{S}))} < \min_{x_j \in \mathbf{M} \setminus \mathbf{M}_1} \frac{w_j r^W - \frac{|\langle \overline{x}_j, \mathbb{P}_{\overline{\mathcal{S}}}(\nu^*) \rangle|}{\|\mathbb{P}_{\overline{\mathcal{S}}}(\nu^*)\|_2}}{\|\overline{x}_j\|_2}, \quad \text{i.e.,}$$

$$\text{dist}(y, \mathcal{S}) < \|\overline{y}\|_2 \frac{\min\limits_{x_j \in \mathbf{M} \setminus \mathbf{M}_1} \frac{w_j r^W - \frac{|\langle \overline{x}_j, \mathbb{P}_{\overline{\mathcal{S}}}(\nu^*) \rangle|}{\|\mathbb{P}_{\overline{\mathcal{S}}}(\nu^*)\|_2}}{\|\overline{x}_j\|_2}}{1 + \min\limits_{x_j \in \mathbf{M} \setminus \mathbf{M}_1} \frac{w_j r^W - \frac{|\langle \overline{x}_j, \mathbb{P}_{\overline{\mathcal{S}}}(\nu^*) \rangle|}{\|\mathbb{P}_{\overline{\mathcal{S}}}(\nu^*)\|_2}}{\|\overline{x}_j\|_2}}, \quad \text{(A.21)}$$

then there exists a non-empty interval $(\lambda^l, \lambda^u) := \left( \frac{1}{r^W(\|\overline{y}\|_2 - \text{dist}(y, \mathcal{S}))}, \min\limits_{x_j \in \mathbf{M} \setminus \mathbf{M}_1} \frac{w_j r^W - \frac{|\langle \overline{x}_j, \mathbb{P}_{\overline{\mathcal{S}}}(\nu^*) \rangle|}{\|\mathbb{P}_{\overline{\mathcal{S}}}(\nu^*)\|_2}}{r^W \text{dist}(y, \mathcal{S}) \|\overline{x}_j\|_2} \right)$.

And for any $\lambda \in (\lambda^l, \lambda^u)$, every optimal solution to (WMC) is manifold preserving. $\qquad \square$

## A.2  Proof of Proposition 1

*Proof of Proposition 1.* To prove Proposition 1, we need to derive conditions (based on an optimal solution to the reduced dual 2.6) for which any optimal solution to (WMC) will be manifold preserving. We do so by (i) defining conditions on a primal-dual feasible solutions to (WMC) and its dual (2.3) that ensure manifold preserving optimal solution to (WMC) (see Lemma 4), and then (ii) constructing such primal-dual feasible pair to (WMC) from the optimal primal-dual solution to the reduced problem (2.5) and its dual (2.6).

The following lemma provides conditions for any optimal solution of (WMC) to be manifold preserving. A similar result was given in [62] for linear sparse clustering problem. In this work, we have adapted [62, Lemma 12] for our proposed model (WMC).

**Lemma 4.** *Let* $(c, \overline{e}, \nu)$ *be a primal-dual feasible solution to* (WMC) *and its dual* (2.3) *such that* $c$ *has support* $B$ *where* $B \subseteq A \subseteq \{1, \ldots, N\}$, *and the dual feasible solution satisfies:*

1. $(W^{-T} \overline{X}^T)_B \nu = \text{sgn}((Wc)_B)$,

2. $\nu = \lambda \overline{e}$,

3. $\|\overline{X}_{A \cap B^c}^T \nu\|_{\infty, W^{-1}} \leq 1$,

4. $\|\overline{X}_{A^c}^T \nu\|_{\infty, W^{-1}} < 1$.

*Then every optimal solution $(c^{**}, \overline{e}^{**}, \nu^{**})$ to* (WMC) *and its dual* (2.3) *satisfies $c_{A^c}^{**} = 0$.*

*Proof.* $(c^{**}, e^{(1)**}, e^{(2)**})$ be an optimal solution to (WMC). We then have

$$\|c^{**}\|_{1,W} + \frac{\lambda}{2}\|\overline{e}^{**}\|_2^2 = \|c_B^{**}\|_{1,W} + \|c_{A\cap B^c}^{**}\|_{1,W} + \|c_{A^c}^{**}\|_{1,W} + \frac{\lambda}{2}\|\overline{e}^{**}\|_2^2$$
$$\geq \|c_B\|_{1,W} + \langle \mathrm{sgn}((Wc)_B), (Wc^{**})_B - (Wc)_B\rangle \qquad (A.22)$$
$$+ \|c_{A\cap B^c}^{**}\|_{1,W} + \|c_{A^c}^{**}\|_{1,W} + \frac{\lambda}{2}\|\overline{e}^{**}\|_2^2.$$

We now determine a lower bound on $\|\overline{e}^{**}\|_2^2$ from the following proposition.

**Proposition 4.** *For any $\gamma > 0$, the function*

$$g(\overline{e}) = \gamma\left(\overline{e}^T\overline{e}^{**} - \frac{1}{2}\overline{e}^T\overline{e}\right) \qquad (A.23)$$

*has a unique maximizer $\overline{e}^{**}$ and $g(\overline{e}^{**}) = \frac{\gamma}{2}\|\overline{e}^{**}\|_2^2$.*

Setting $\gamma = \lambda$, we have

$$\frac{\lambda}{2}\|\overline{e}^{**}\|_2^2 = g(\overline{e}^{**})$$
$$\geq g(\overline{e})$$
$$= \lambda\left(\overline{e}^T\overline{e}^{**} - \frac{1}{2}\overline{e}^T\overline{e}\right) \qquad (A.24)$$
$$= \frac{\lambda}{2}\|\overline{e}\|_2^2 + \langle \lambda\overline{e}, \overline{e}^{**} - \overline{e}\rangle.$$

Also, from condition 1 in the lemma, we have

$$\langle \mathrm{sgn}((Wc)_B), (Wc^{**})_B - (Wc)_B\rangle = \langle (W^{-T}\overline{X}^T)_B\nu, (Wc^{**})_B - (Wc)_B\rangle$$
$$= \langle \nu, \overline{X}_B(c_B^{**} - c_B)\rangle$$
$$= \langle \nu, \overline{X}(c^{**} - c)\rangle - \langle \nu, \overline{X}_{A\cap B^c}(c_{A\cap B^c}^{**})\rangle - \langle \nu, \overline{X}_{A^c}(c_{A^c}^{**})\rangle. \qquad (A.25)$$

Combining (A.22), (A.24), and (A.25), we have

$$\|c^{**}\|_{1,W} + \frac{\lambda}{2}\|\overline{e}^{**}\|_2^2 = \|c_B\|_{1,W} + \frac{\lambda}{2}\|\overline{e}\|_2^2$$
$$+ \|c_{A\cap B^c}^{**}\|_{1,W} - \langle \nu, \overline{X}_{A\cap B^c}(c_{A\cap B^c}^{**})\rangle$$
$$+ \|c_{A^c}^{**}\|_{1,W} - \langle \nu, \overline{X}_{A^c}(c_{A^c}^{**})\rangle \qquad (A.26)$$
$$+ \langle \nu, \overline{X}(c^{**} - c)\rangle + \langle \lambda\overline{e}, \overline{e}^{**} - \overline{e}\rangle.$$

Now, from condition 2 in the lemma, we have that

$$\langle \nu, \overline{X}(c^{**} - c)\rangle + \langle \lambda\overline{e}, \overline{e}^{**} - \overline{e}\rangle = \langle \nu, \overline{X}(c^{**} - c) + \overline{e}^{**} - \overline{e}\rangle = 0, \qquad (A.27)$$

where the last inequality follows since $(c, \overline{e})$ and $(c^{**}, \overline{e}^{**})$ are both feasible to (WMC), and so, $\overline{y} = \overline{X}c + \overline{e} = \overline{X}c^{**} + \overline{e}^{**}$. Furthermore, we have that

$$\|c_{A\cap B^c}^{**}\|_{1,W} - \langle \nu, \overline{X}_{A\cap B^c}(c_{A\cap B^c}^{**})\rangle = \|c_{A\cap B^c}^{**}\|_{1,W} - \langle (W^{-T}\overline{X}^T)_{A\cap B^c}\nu, ((Wc^{**})_{A\cap B^c})\rangle$$
$$\geq \|c_{A\cap B^c}^{**}\|_{1,W} - \|\overline{X}_{A\cap B^c}^T\nu\|_{\infty,W^{-1}}\|c_{A\cap B^c}^{**}\|_{1,W}$$
$$\geq 0, \qquad (A.28)$$

where the last inequality follows from condition 3 in the lemma. Similarly, we have that

$$\|c_{A^c}^{**}\|_{1,W} - \langle \nu, \overline{X}_{A^c}(c_{A^c}^{**})\rangle = \|c_{A^c}^{**}\|_{1,W} - \langle (W^{-T}\overline{X}^T)_{A^c}\nu, ((Wc^{**})_{A^c})\rangle$$
$$\geq \|c_{A^c}^{**}\|_{1,W}(1 - \|\overline{X}_{A^c}^T\nu\|_{\infty,W^{-1}}). \qquad (A.29)$$

Combining (A.26), (A.27), (A.28), and (A.29), and using the fact $B$ is the support of $c$, we have

$$\|c^{**}\|_{1,W} + \frac{\lambda}{2}\|\overline{e}^{**}\|_2^2 \geq \|c\|_{1,W} + \frac{\lambda}{2}\|\overline{e}\|_2^2 + \|c_{A^c}^{**}\|_{1,W}(1 - \|\overline{X}_{A^c}^T\nu\|_{\infty,W^{-1}}). \qquad \text{(A.30)}$$

From condition 4 in the lemma, we know that $\|\overline{X}_{A^c}^T\nu\|_{\infty,W^{-1}} < 1$. Since $(c^{**}, \overline{e}^{**}, \nu^{**})$ is optimal, this implies that $\|c_{A^c}^{**}\|_{1,W} = 0$, and therefore, $(Wc^{**})_{A^c} = 0$, proving the result. $\qquad \square$

Lemma 4 provides conditions for any optimal solution of (WMC) to be manifold preserving. Now, we construct a feasible solution $(c, \overline{e}, \nu)$ to (WMC) that satisfies the conditions given in Lemma 4 from an optimal solution $(c^*, \overline{e}^*, \nu^*)$ to the reduced problem (2.5) and its dual (2.6) as follows:

$$c_j = \begin{cases} c_j^* & \text{for } j \text{ such that } x_j \in \mathbf{M}_1 \cap \mathbf{N} \\ 0 & \text{otherwise,} \end{cases} \qquad \text{(A.31)}$$

$$\overline{e}^* = \overline{y} - \overline{X}_{\overline{\mathcal{S}}}c^* = \overline{y} - \overline{X}c = \overline{e}, \quad \text{and} \qquad \text{(A.32)}$$

$$\nu = \nu^*. \qquad \text{(A.33)}$$

We observe that $(c, \overline{e}, \nu)$ is feasible to (WMC) by construction. Let $T$ be the set of indices of data samples in the set $\mathbf{M}_1 \cap \mathbf{N}$, and let $B$ be the support of $c^*$. From KKT conditions for the reduced problem (2.5) and its dual (2.6), we have

$$\nu^* = \lambda\overline{e}^* = \nu = \lambda\overline{e}, \quad \text{and} \qquad \text{(A.34)}$$

$$W^{-T}\overline{X}_{\overline{\mathcal{S}}}^T\nu^* = \text{sgn}(Wc^*) \equiv (W^{-T}\overline{X}^T)_B\nu^* = \text{sgn}((Wc)_B). \qquad \text{(A.35)}$$

Thus, $(c, \overline{e}, \nu)$ satisfies the first two conditions in Lemma 4. Moreover, we have that

$$\|\overline{X}_{A\cap B^c}^T\nu\|_{\infty,W^{-1}} \leq \|\overline{X}_A^T\nu\|_{\infty,W^{-1}} = \|\overline{X}_{\overline{\mathcal{S}}}^T\nu\|_{\infty,W^{-1}} \leq 1, \qquad \text{(A.36)}$$

which shows that condition 3 from Lemma 4 is satisfied for $(c, \overline{e}, \nu)$. Finally, we observe that condition 4 in Lemma 4 can be equivalently written as

$$|\langle \overline{x}_j, \nu^* \rangle| < w_j \quad \forall x_j \in \mathbf{M} \setminus \mathbf{M}_1. \qquad \text{(A.37)}$$

Thus, we have that if (A.37) is satisfied, then all conditions in Lemma 4 are satisfied, and every optimal solution to (WMC) is manifold preserving. $\qquad \square$

## A.3 Proof of Lemma 2

*Proof of Lemma 2.* We prove the result using geometric properties of the manifold and location of the data samples. In the proof, we will introduce several new quantities and notations.

We define the curvature of the manifold $\mathcal{M}_1$ at $y$ to be $\kappa$. From the definition of the curvature of the smooth manifold, we have that $1/\kappa$ is the radius of the largest internal tangent sphere to the manifold $\mathcal{M}_1$ at $y$. Let the tangent circle be denoted by $\mathcal{T}$ and let $z$ be the center of the $(D-1)$-dimensional internal tangent sphere. Consider a point of intersection of the sphere $\mathcal{B}$ with $\mathcal{T}$ and denote it by $u$. Next, we project the point $u$ on to the line connecting $y$ and $z$, and denote the projected point by $v$.

We then have that the points $z$, $u$ and $v$ form vertices of a right-angled triangle, and we have

$$\|z - u\|_2^2 = \|z - v\|_2^2 + \|v - u\|_2^2. \qquad \text{(A.38)}$$

We now determine the bounds on each of the quantities in (A.38).

Since $z$ is the center of the sphere $\mathcal{T}$ of radius $1/\kappa$, and $u$ lies on the surface of the sphere, we have

$$\|z - u\|_2^2 = \frac{1}{\kappa^2}. \qquad \text{(A.39)}$$

Now consider the three points $y$, $u$ and $v$. Since $v$ is the projection of the point $u$ on to the line connecting $y$ and $z$, we have that the three points $y$, $u$ and $v$ also form a right-angled triangle. Thus, we have that

$$\|u - v\|_2^2 \leq \|u - y\|_2^2 = \zeta^2, \qquad \text{(A.40)}$$

where the last inequality follows since $u$ lies on the boundary of the sphere $\mathcal{B}$ with radius $\zeta$ and centered at $y$.

Finally, we bound $\|z - v\|_2$. Since $v$ is the projection of point $u$ on to the line connecting $y$ and $z$, we have that $z$, $v$ and $y$ collinear and they lie on a single line connecting $y$ and $z$. We, therefore, have that $\|z - v\|_2 = \|y - z\|_2 - \|v - y\|_2$. Furthermore, since $z$ is the center of the tangent sphere to the manifold $\mathcal{M}_1$ at $y$, we have that $\|y - z\|_2 = \frac{1}{\kappa}$. Thus,

$$\|z - v\|_2 = \frac{1}{\kappa} - \|v - y\|_2. \tag{A.41}$$

**Proposition 5.** *For the point $v$ as defined above, we have dist$(y, \mathcal{S}) \leq \|y - v\|_2$.*

*Proof.* Let $\mathcal{BT}$ be the set consisting of $u$ and additional $d_1$ unique points of intersection of the sphere $\mathcal{B}$ with $\mathcal{T}$. We have that $\dim(\text{aff}\{x : x \in \mathcal{BT}\}) \leq d_1$. Since each point in the set $\mathcal{BT}$ lies on the boundaries of both $\mathcal{B}$ and $\mathcal{T}$, we have that each point $x \in \mathcal{BT}$ is equidistant from both $y$ and $z$. Thus, $u$ and every point $x \in \mathcal{BT}$ projects onto the same point ($v$) on the line connecting $y$ and $z$. Furthermore, we have that

$$\|y - v\|_2 = \text{dist}(y, \text{aff}\{x : x \in \mathcal{BT}\}), \tag{A.42}$$

i.e., $v$ is the projection of $y$ onto the affine subspace $\text{aff}\{x : x \in \mathcal{BT}\}$.

From the definition of the inner tangent circle $\mathcal{T}$, we have that the affine subspace $\text{aff}\{x : x \in \mathcal{BT}\}$ will intersect with manifold $\mathcal{M}_1$ at points outside the sphere $\mathcal{B}$. Thus, $\text{aff}\{x : x \in \mathcal{BT}\}$ is an affine subspace of dimension $\leq d_1$ spanned by points not in $\mathcal{B}$. From the definition of $\mathcal{S}$ (see Definition 3) and Assumption 2, we note that $\mathcal{S}$ is the affine subspace (of dimension $d_1$) nearest to $y$, and spanned by the points inside or on the sphere $\mathcal{B}$. Thus, we have that

$$\text{dist}(y, \mathcal{S}) \leq \text{dist}(y, \text{aff}\{x : x \in \mathcal{BT}\}). \tag{A.43}$$

Combining (A.42) and (A.43) proves the result. $\qquad\square$

From (A.41) and Proposition 5, we bound $\|z - v\|_2$ as

$$\|z - v\|_2 \leq \frac{1}{\kappa} - \text{dist}(y, \mathcal{S}). \tag{A.44}$$

Combining (A.38), (A.39), (A.40) and (A.44), we have that

$$\frac{1}{\kappa^2} \leq \zeta^2 + \left(\frac{1}{\kappa} - \text{dist}(y, \mathcal{S})\right)^2, \tag{A.45}$$

or equivalently,

$$\left(\frac{1}{\kappa} - \text{dist}(y, \mathcal{S})\right)^2 \geq \frac{1}{\kappa^2} - \zeta^2. \tag{A.46}$$

Since $\text{reach}(\mathcal{M}_1) \geq \zeta$, and from Definition 8, $\frac{1}{\kappa} \geq \text{reach}(\mathcal{M}_1)$, we have that $\frac{1}{\kappa} \geq \zeta$, and thus,

$$\frac{1}{\kappa} - \text{dist}(y, \mathcal{S}) \geq \sqrt{\frac{1}{\kappa^2} - \zeta^2}. \tag{A.47}$$

Rearranging the terms proves the result. $\qquad\square$

# B  Details of Experiments on Real Data

## B.1  CIFAR Datasets and CLIP feature extraction

**CIFAR Datasets.**  CIFAR consists of 50,000 training and 10,000 test color images of size 32x32. They are equally divided into 10, 20, and 100 classes resulting in CIFAR-10, CIFAR-20, and CIFAR-100 datasets respectively. The 20 classes in CIFAR-20 are obtained by merging 100 classes in CIFAR-100 into disjoint groups.

**CLIP Pre-Training Model.**  Contrastive Language-Image Pre-training (CLIP) [63] is an efficient tool for image representation learning by jointly training an image encoder and a text encoder to *correctly* predict the image-text pairings. We use ViT-L/14 [66] architecture for image encoder. ViT-L/14 refers to 'Large' Vision Transfomer model with input images divided into patches of size 14x14. See [67] for more information on the definition of 'Large' models. We use the features extracted from images in CIFAR dataset by CLIP pre-training model that uses ViT-L/14 architecture[*].

---

[*]We use the CLIP package available at https://github.com/openai/CLIP?tab=readme-ov-file

## B.2 Details of Experimental Setup

We include details of experiments performed in Section 3.2. Note that, the results reported in Table 1 for SSC, EnSC, CPP and TEMI methods are taken from [56]. Please refer to [56, Section 4] for experimental details.

**L-WMC and E-WMC.** We perform the steps in Algorithm 1 for clustering data using L-WMC and E-WMC. Details of pre-training using CLIP (Step 1 of Algorithm 1) are provided in Appendix B.1. To construct the representation matrix (Step 2), we define the models L-WMC and E-WMC for each input data sample with the distance matrix $W$ set as given in Section 3.2.2. We use an efficient active-set solver given in [18] to solve the optimization problem. The experiments are performed on a machine with Intel(R) Xeon(R) Gold 6130 CPU operating at 2.10 GHz frequency and with 37 GB RAM. We have provided our code in the Supplementary material.

We use grid search over the following parameter values: $\eta \in \{1, 20, 100, 400\}$ and $\lambda \in \{20, 50\} \times \lambda_0$, where $\lambda_0$ is the smallest value of $\lambda$ that generates a non-trivial (non-zero) solution. We report the best accuracy results in Table 1. Furthermore, Table 2 provides the values of the parameters $\lambda$ and $\eta$ corresponding to the clustering results reported in rows 1 (L-WMC) and 2 (E-WMC) in Table 1.

| Method/ Parameter values | CIFAR-10 | | CIFAR-20 | | CIFAR-100 | |
|---|---|---|---|---|---|---|
| | $\lambda$ | $\eta$ | $\lambda$ | $\eta$ | $\lambda$ | $\eta$ |
| L-WMC | 50 | 20 | 50 | 400 | 50 | 20 |
| E-WMC | 20 | 400 | 20 | 400 | 50 | 400 |

Table 2: Values of the parameters corresponding to the clustering results reported in Table 1 for L-WMC and E-WMC.

**SMCE.** The MATLAB code for the ADMM algorithm was provided by the authors of [58]. We implemented the ADMM algorithm that solves SMCE [58] in Python. We refer to this method as SMCE in this paper. As discussed in Section 3.2.2, we apply SMCE to pre-trained data generated using CLIP. To generate the representation matrix, SMCE solves the following optimization problem for each input sample $x_i$,

$$\min_{c_i} \gamma \|W_i c_i\|_1 + \frac{1}{2} \|Q_i c_i\|_2^2 \quad \text{s.t.} \quad \mathbf{1}^T c_i = 1, \tag{B.1}$$

where $W_i$ is a diagonal matrix whose $j$-th entry $[W_i]_{jj} = w_j = \frac{\|x_j - x_i\|_2}{\sum_{k \neq i} \|x_k - x_i\|_2}$, and $Q_i = \left[\frac{x_j - x_i}{\|x_j - x_i\|_2}\right]_{j \neq i}$. Note that the matrix $W_i$ is the same as the matrix $W$ defined for our model L-WMC. Furthermore, since the $\ell_1$ does not select points that are very far from the given data point $x_i$, [58] consider only $K$ nearest neighbors of the point $x_i$ in the algorithm where $K$ is chosen *a priori*. We use grid search to choose the values of the hyperparameters $\gamma$ and $K$: $\gamma \in \{10, 20, 50\}$ and $K \in \{10, 20, 50, 100\}$. Table 1 provides the best clustering result achieved by SMCE for each of the three datasets. In Table 3, we provide the parameter values corresponding to the clustering results reported for SMCE in Table 1.

| Method/ Parameter values | CIFAR-10 | | CIFAR-20 | | CIFAR-100 | |
|---|---|---|---|---|---|---|
| | $\gamma$ | $K$ | $\gamma$ | $K$ | $\gamma$ | $K$ |
| SMCE | 10 | 20 | 50 | 20 | 20 | 20 |

Table 3: Values of the parameters corresponding to the clustering results reported in Table 1 for SMCE.

## B.3 Constructing affinity

The first step in using spectral clustering to cluster the input data samples is to construct an affinity matrix that indicates pairwise similarities between the data samples. Ideally, the data samples from the same cluster should be highly connected while those from different clusters should have no connections. For the models based on self-expressiveness property, the construction of the affinity

matrix involves post-processing the representation matrix $C$. Several post-processing strategies have been used in literature to construct a symmetric affinity matrix from the representation matrix. We describe three commonly used strategies below.

- **Normalize (N)**: This strategy of constructing affinity matrix $A$ involves normalizing each column of the representation matrix $C$ to have unit $\ell_2$ norm. This strategy can be helpful when the input data samples have different norms.
- **Symmetric (S)**: This is one of the commonly used strategies to construct the affinity matrix $A$ as $A = \frac{1}{2}(|C| + |C|^T)$.
- $k$-**NN (K)**: This strategy involves finding $k$ nearest neighbors of each data sample. Note that, the representation matrix can have dense connections. When $k$-NN strategy is applied to each column of $C$, only the $k$ largest entries in the column are preserved while the rest are set to 0. Thus, the affinity matrix resulting from this strategy will have at most $k$ non-zero entries in each column. This strategy can be helpful when there are several nonzero entries in $C$ of very small magnitude.

We use $N$, $S$, $K$ to denote the three strategies from now on. Moreover, we use $SNK$ to denote that 'symmetric', 'normalize' and '$k$-NN' strategies are used one after the other (and in that order) on the representation matrix $C$ to construct the affinity matrix. Since no universally best strategy exists, we construct an affinity matrix by defining all permutations of all possible subsets of the three strategies described above. Thus, there are 15 possible ways to construct an affinity by applying the following strategies on $C$: $N$, $S$, $K$, $NK$, $KN$, $SN$, $NS$, $SK$, $KS$, $SNK$, $SKN$, $KSN$, $KNS$, $NSK$, $NKS$. After constructing the affinity matrix using each strategy or a combination of strategies, we use $k$-means to cluster the data.

For the clustering results reported in Table 1 for L-WMC, E-WMC, and SMCE, we provide the affinity construction strategy corresponding to these results in Table 4.

| Method/Dataset | CIFAR-10 | CIFAR-20 | CIFAR-100 |
|---|---|---|---|
| L-WMC | N | SNK | N |
| E-WMC | NK | SNK | N |
| SMCE | S | SNK | NS |

Table 4: Strategy used to construct affinity from the representation matrix corresponding to the clustering results reported in Table 1 for L-WMC and E-WMC.

We observed that the CLIP feature vectors for each image in the CIFAR datasets have different $\ell_2$ norm. Thus, the columns of the resulting representation matrix $C$ also have different sizes ($\ell_2$ norm). In this case, it seems intuitive to normalize the columns of $C$ while constructing the affinity. Indeed, from Table 4, we notice that the best clustering accuracy result was observed when the affinity construction involved normalizing the representation matrix $C$.

## C  Review of Manifold Clustering Methods with Theoretical Guarantees

**Distance between Tangent Spaces**    The work of [68] provides an algorithm that i) fits an affine subspace to a local neighborhood of each point in the dataset and ii) computes the similarity between two points based on the distance between the two subspaces as well as that between the two points. Further, they provide an upper bound on Euclidean distances between data samples from different clusters to achieve perfect clustering with high probability. Interestingly, the above was later extended to allow for a faster computation based on prototypes [69] and the ambient space being a Riemannian manifold [70, 71]. However, their theoretically guaranteed algorithms require knowledge of the intrinsic dimension of each manifold (or implicitly a model selection parameter to estimate the dimension) which is in practice typically not known apriori and hard to estimate.

**Longest-Leg Path Distance**    The work of [72] provides an algorithm that computes distance in a different way: for every two points $x, y$ in a given set of data points, the distance between $x, y$ is the minimum over all paths connecting $x, y$ of the longest leg in the path. Then, the similarity can be computed by applying a kernel function to the distances. They provide theoretical guarantees that allow for a general number of manifolds, as well as algorithms that approximate the distances hierarchically.

Nonetheless, the approaches have been only tested on smaller-scale datasets (i.e., either ambient dimension, or number of data samples, or number of clusters being small, while this paper shows both theoretical guarantees as well as clustering performance on large-scale datasets of natural images. A direction of independent interest to the community is to understand the computational picture of these methods on such datasets, which is not pursued by this paper.

