# OpenReview forum: "Geometric Analysis of Nonlinear Manifold Clustering"
_NeurIPS.cc/2024/Conference — NeurIPS 2024 poster_

### Official Review · Reviewer_Mh3Q · 2024-06-18

**Soundness:** 4
**Presentation:** 4
**Contribution:** 3
**Rating:** 7
**Confidence:** 3

**Summary:**

The authors present a novel approach for nonlinear manifold clustering that comes with both a solid theoretical background and provable guarantees as well as some experiments indicating practical applicability.

**Strengths:**

The paper is well-written and in this reviewers opinion fits very well into Neurips. The studied problem is relevant for many applications. The proofs that this reviewer has checked appear to be correct.

**Weaknesses:**

The main downside I see is that the proposed method relies on many crucial hyperparameters, and the authors could go into a bit more detail into how these parameters should be selected in practice.

**Questions:**

What is the reason for choosing the values in the grid search for the hyperparameters in the way you did?

**Limitations:**

The authors discuss some limitations in Section 4 of the manuscript.

---

> ### Author Rebuttal · Authors · 2024-08-07
>
> We are grateful for your positive evaluation of our work, especially for acknowledging that *“the paper … in this reviewers opinion fits very well into Neurips”*. Thank you for the constructive comments, which we address below.
>
> **How to choose hyperparameters:**
>
> There are three hyperparameters of the method, $W$, $\eta$, and $\lambda$, which we discuss below.
> - Choosing $W$: We define the weight matrix such that each entry $(i,j)$ is proportional to (a) the distance between data samples $x_i$ and $x_j$ (see model L-WMC), (b) the exponential of the distance between $x_i$ and $x_j$ (see model E-WMC).
> - Choosing $\eta$: We perform a grid search over multiple lambda values, and the following grid values for eta: (a) 1 (b) empirical mean of $||x||2$, (c) empirical mean of $||x||_2^2$.
> - Choosing $\lambda$: Please kindly refer to the response to Reviewer XZ5T.
>
> We hope this alleviates your concern; if not, we are happy to engage with you during the discussion session. Thank you for the question!
>
> Best Regards,
>
> Authors of Submission 17809

---

> > ### Comment · Reviewer_Mh3Q · 2024-08-08
> >
> > I would like to thank the authors for their clarifying rebuttal.

---

### Official Review · Reviewer_yHKx · 2024-07-12

**Soundness:** 3
**Presentation:** 3
**Contribution:** 3
**Rating:** 6
**Confidence:** 4

**Summary:**

This paper attempts to present a theoretical analysis for a slightly modified sparse manifold clustering method, showing that under some condition on the data distribution, the separation between different manifolds and the curvature of the manifold, the optimal solution is point-wise subspace preserving and thus implies manifold preserving. In addition, the range of the hyperparameter $\lambda$ is derived and can be nonempty (if the density of the data in the manifold is high enough). Empirical evaluations are conducted on both synthetic data (to validate the theoretical results) and CIFAR10/20/100 with pretrain CLIP feature.

**Strengths:**

+ The paper derives the theoretical  condition to guarantee the manifold preserving solution for the modified sparse manifold clustering model (which is based on a sparse self-expressive model).

+ The paper provides clear geometric interpretations on the derived theoretical conditions, also the curvature of the manifold is connected to the density of the data per manifold.

**Weaknesses:**

1. The formulation in Eq.(1.1) is nothing but a trivial modification from SMCE [57].

- In SMCE, it also adopts a locality weighted $\ell_1$ norm based regulizer. To be specific, the so-called L-WMC uses a linerly weighting scheme (as in L295), which is exactly the same as that in SMCE (where the weight matrix is $Q_i$).  So, what is the essential difference from the weighted sparse model in SMCE?

2. An extra constraint $1^\top c =1$ is imposed on the objective function as a penalty term. Why such a constraint $1^\top c =1$ is imposed on the objective as a penalty? Is there any geometric interpretation?

3. Compared to the strict constraint, what is the reason to relax it as a penalty term?  Moreover, is the parameter $\eta$ set as very large? If not, why not simply impose an affine constraint to have an affine combination?

4. In Eq. (WMC), a homogenization of the data is used, rather than using the raw data. So, are the input data properly normalized or not? Why or why not? Since that $\eta$ should be as large as possible, is that reasonable to appending a very large component on the data vector $x$?

5. Since that the data points are neighboring points and the data are not normalized (the reviewer did find out the normalization step), what can we expect the size of the inradius?

6. The experiments are insufficient and less convincing.

- The performance comparison is neighther complete nor fair enough.
Since that the so-called WMC is similar to SMCE (as discussed in L55-L66), SMCE is the most important baseline method. However, the experimental results listed in Table 1 don't include SMCE. Thus, the empirical evaluation is less convincing.

- For the results of EnSC and SSC listed in Table 1, is there some suitable postprocessing adopted as that in L-WMC or E-WMC? How much the performance improvement is coming from the spetial postprocessing step? What about the performance when the baseline methods also adopt a proper postprocessing?

7. In L42: the piled prior works are deep subspace clustering methods, none of them is for nonlinear manifold clustering method. Also, it is not a good practice to pile many citations in a single bracket.

**Questions:**

1. The formulation in Eq.(1.1) is a modification from SMCE [57]. What is the essential difference from the weighted sparse model in SMCE?

2. Why such a constraint $1^\top c =1$ is imposed on the objective as a penalty? Is there any geometric interpretation?

3. Compared to the strict constraint, what is the reason to relax it as a penalty term?  Moreover, is the parameter $\eta$ set as very large? If not, why not simply impose an affine constraint to have an affine combination?

4. In Eq. (WMC), a homogenization of the data is used, rather than using the raw data. So, are the input data properly normalized or not? Why or why not? Since that $\eta$ should be as large as possible, is that reasonable to appending a very large component on the data vector $x$?

5. Since that the data points are neighboring points and the data are not normalized (the reviewer did find out the normalization step), what can we expect the size of the inradius?

6. The experiments are insufficient and less convincing.

- The performance comparison is neighther complete nor fair enough. Since that the so-called WMC is similar to SMCE (as discussed in L55-L66), SMCE is the most important baseline method. However, the experimental results listed in Table 1 don't include SMCE. Thus, the empirical evaluation is less convincing.

- For the results of EnSC and SSC listed in Table 1, is there some suitable postprocessing adopted as that in L-WMC or E-WMC? How much the performance improvement is coming from the spetial postprocessing step? What about the performance when the baseline methods also adopt a proper postprocessing?

7. In L42: the piled prior works are deep subspace clustering methods, none of them is for nonlinear manifold clustering method. Also, it is not a good practice to pile many citations in a single bracket.

**Limitations:**

Yes

---

> ### Author Rebuttal · Authors · 2024-08-07
>
> Thank you for your time in reviewing our paper. We are happy to address your comments below.
>
> **(W1, W2, W3, W6) Comparison with SMCE**
>
> It was questioned how our method differs from SMCE, why such a difference is needed, and if the difference is significant.
> - We agree that our formulation (1.1) differs from SMCE by penalizing the affine constraint.
> - The modification is motivated by the goal of providing theoretical guarantees of non-linear manifold clustering methods.
>     - With the affine constraint in the SMCE model, we are unable to provide an upper bound on $\lambda$. This prompted us, by necessity, to relax the constraint as a penalization leading to (1.1), which allows us to derive the bounds and theorems in section 2.
> - The focus and contributions of this paper are therefore two-fold:
>     - Providing geometric conditions that guarantee a manifold preserving solution of (1.1) in terms of curvature, sampling density, separation between the manifold;
>     - Showing that (1.1) performs comparably to the state-of-the-art manifold clustering algorithms.
>
>
> **(W2) Geometric interpretation of the constraint $1^Tc = 1$ or its penalty version**
> - The affine constraint in SMCE is motivated by the fact that a manifold can be locally approximated by an affine subspace.
> - Making it a soft penalty with multiplier $\eta$ is equivalent to homogenizing the data with homogenization constant $\sqrt{\eta}$, i.e., (WMC) is equivalent to (1.1).
>
> We will include the clarification in the preamble of section 2, thanks for the comment!
>
> **(W3) Is the parameter $\eta$ set as very large?**
>
> In our experiments, the parameter $\eta$ is not chosen to be very large. We chose 3 different values of $\eta$ proportional to: (a) empirical mean of $||x||^2$, (b) empirical mean of $||x||$, and (c) the value 1.
>
> **(W4) In Eq. (WMC), a homogenization of the data is used, rather than using the raw data. (Part 1) So, are the input data properly normalized or not? Why or why not? (Part 2) Since that $\eta$ should be as large as possible, is that reasonable to appending a very large component on the data vector $x$? (Part 3)**
> - Part 1: Eq. (WMC) is equivalent to the objective (1.1), i.e., objective with affine constraint penalization, as we explained in the response above.
> - Part 2: We did not normalize the input data. As a result, we see that the quantities in our geometric result (Lemma 1) depend on the norm of the data samples. We observe that the same analysis can be used for normalized data samples, which results in a special case.
> - Part 3: Great question!
>     - Making $\eta$ as large as possible has two effects: on the one hand, it helps enforce the affine constraint. On the other hand, it diminishes the importance of the reconstruction error term in (1.1); a different perspective is that the large $\eta$ dominates the computation of the Euclidean norm of the homogenized points.
>     - So it is not reasonable to append a large $\eta$ to the data sample $x$. Thus, rather than choosing an arbitrarily large value for $\eta$ in our experiments, we choose $\eta$ as described in the response of W3.
>
> **(W5) Since that the data points are neighboring points and the data are not normalized (the reviewer did find out the normalization step), what can we expect the size of the inradius?**
>
> The size of the inradius does depend on the size of the data samples. If the norm of few (or all) data samples scales up, then the inradius will also scale up. However, note that this relationship is not linear. Furthermore, we believe that simply scaling up or down all the data samples by a constant will not have a significant impact on the theoretical results (Lemma 1) since other quantities (apart from inradius) involved in the result also change accordingly.
>
> **(W6) The experiments are insufficient and less convincing.**
>
> **1. The performance comparison is neighther complete nor fair enough. Since that the so-called WMC is similar to SMCE (as discussed in L55-L66), SMCE is the most important baseline method. However, the experimental results listed in Table 1 don't include SMCE. Thus, the empirical evaluation is less convincing.**
>
> The goal of our experiments was to compare the performance of WMC with current state-of-the-art methods and show that WMC performs only slightly worse than these methods while also providing a theoretical understanding of the model.
>
> **2. For the results of EnSC and SSC listed in Table 1, is there some suitable postprocessing adopted as that in L-WMC or E-WMC? How much the performance improvement is coming from the spetial postprocessing step? What about the performance when the baseline methods also adopt a proper postprocessing?**
>
> EnSC and SSC did adopt/use post-processing; in particular, they use the symmetric normalization strategy (“S” in Appendix B.4). We also used the post-processing strategy described in Appendix B.4 when SSC method was used to cluser points in CIFAR-10 dataset. The clustering accuracy was observed to 86.4% (compared to 96.07% for our method).
>
> **(W7) In L42: the piled prior works are deep subspace clustering methods, none of them is for nonlinear manifold clustering method. Also, it is not a good practice to pile many citations in a single bracket.**
>
> We thank the reviewer for the suggestion, which we will certainly adopt for the final submission.
>
>
> Best Regards,
>
> Authors of Submission 17809

---

> > ### Comment · Reviewer_yHKx · 2024-08-11
> > **Responses to rebuttal**
> >
> > Thanks for the point-to-point respondes in the rebuttal.
> >
> > Since the parameter $\eta$ cannot set to large, it is less reasonable to set it to infinity (in the dicussion at the end of Section 1).
> >
> > The reviewer did not satisfy the responses for the insufficiency in experiments. The high clustering accuracy seems owning to the pre-trained feature via CLIP, not the slightly modification from SMCE.
> > The reviewer appreciates the established theoretical anaysis for understanding a modified sparse manifold clustering algorithm, but not the empirical evaluation is weak.
> > Had well tuned the parameters and conducted fair experiments, whanever EnSC or SMCE, could yield the similar high performance. A solid and convincing empirical evaluation could helps further.

---

> > > ### Author Response · Authors · 2024-08-13
> > >
> > > We greatly appreciate your increased rating and your additional input. Below we take the opportunity to further alleviate your concerns.
> > >
> > > *"Since the parameter $\eta$ cannot set to large, it is less reasonable to set it to infinity”*
> > > - You are right, we will remove the claim in the revised paper. Thank you for the catch!
> > >
> > > *“The high clustering accuracy seems owning to the pre-trained feature via CLIP”*
> > > - All of the methods in Table 1 use CLIP: EnSC, SSC directly cluster on CLIP features; TEMI, CPP used deep networks to learn to refine and cluster CLIP features.
> > > - Promised Revision:  We are motivated by your comment, and in the revision, we will highlight the above in the caption of Table 1, rather than simply saying “when applied to CLIP features”. Thank you!
> > >
> > > *“...not the slightly modification from SMCE.”*
> > > - The modification from SMCE to our formulation WMC is for providing theoretical guarantees of manifold clustering methods, which we tried to be frank about in the rebuttal above (in comparison with SMCE). We provided the current set of experiments to further understand if the very formulation we analyze (WMC) has any impact on real large-scale applications.
> > > - Promised Revision: That being said, your comment is well received: Since WMC is a variant of SMCE, it is interesting to see how SMCE [57] would perform on CIFAR. To our knowledge, the largest dataset that SMCE has been tested on is MNIST as reported by the work of [60].
> > >     - A main barrier that limits us from getting the results during the rebuttal period is the implementation of SMCE: the code provided by the authors of SMCE is in Matlab, and it computes an N x N (i.e., 60,000 x 60,000) dense matrix for computing nearest neighbors, which is expensive. This is even worse if we try to find the optimal parameters.
> > >     - We have been working on implementing SMCE in Python (for a fair comparison with the alternatives) with a modern nearest-neighbor computation. We will gladly include the result in the revised paper.
> > >
> > >
> > > Hopefully this alleviates your concerns. We appreciate your feedback and insights. Please let us know if you have any further concerns or suggestions.
> > >
> > >
> > > Thanks and regards,
> > >
> > > Authors of Submission 17809

---

> > > > ### Comment · Reviewer_yHKx · 2024-08-14
> > > > **Responses to Authors' Responses**
> > > >
> > > > Thanks for the authors' quick responses and the revision promises in the final version.
> > > >
> > > > 1. Normalization or not for the data is an critical issue that worth to discuss a bit in the final version.
> > > >
> > > > 2. Regarding to $\eta$ and the affine constraint
> > > > - Since that the $\eta$ could not be very large, does that mean the affine constraint not helps at the end the day? Could the relaxed affine constraint be drop? If it does not work, it is nothing but a decoration or strawman. Yes, the relaxed constraint is used in the analysis.
> > > >
> > > > - On the other hand, the geometric condition to yield correct nonzero coefficients in the case of linear subspaces is very clear since Sol.&Candes(2012); but the geomtric condition of the correctness guarantee for the case of affine subspaces is rare and less intuitive.  The geometric analysis developed in the submission is built on an interesting concept called nearest affine subspace and a linear subspace of the homogenous embedding. Is there any (intrinsic) connection between the linear subspace of the homogenous embedding and the affine subspace? Is the assumption 2 probably correct (or in what case)? Is there any prior work on subspace clustering in affine setting? Or any theoretical results on the connection from affine geometry (or what else)? A few words on this point would be helpful.
> > > >
> > > > - In addition, it is called an era of large pre-trained model nowadays. But it is also an era calling for transparent "white-box" models. Is there *any* white-box model devoted for nonlinear manifold clustering? It seems that none of the method piled in L42 is.
> > > >
> > > > 3. That sounds quite great if the performance SMCE could be fairly compared to the modified models. As an alternative, it is also okay to compare them on COIL100, which is a standard testbed for manifold clustering algorithms. No need to use pre-trained CILP, no need to use scatter transform or other fancy feature extraction. Use the pixels of the gray scale image as input. Then a fair comparison could be easier.  Lastly, for EnSC, rather than directly citing from prior work, reproducing it on CLIP feature (with parameters are better tuned if possible) would be more convincing.
> > > >
> > > > The responses and the promises could resolve my major concerns. Since that the developed theoretical analysis with the insightful interpretation is clear and solid enough, the reviewer would like to see an acceptance.

---

### Official Review · Reviewer_XZ5T · 2024-07-12

**Soundness:** 4
**Presentation:** 3
**Contribution:** 4
**Rating:** 8
**Confidence:** 4

**Summary:**

The authors addresses the problem of clustering high-dimensional data that lie on multiple, low-dimensional, nonlinear manifolds. They propose a new method that clusters data belonging to a union of nonlinear manifolds, providing geometric conditions for a manifold-preserving representation.
A significant contribution of the paper is the provision of geometric conditions that guarantee the correct clustering of data points on the same manifold. The authors validate their method through experiments on CIFAR datasets, demonstrating competitive performance compared to state-of-the-art methods, although it performs marginally worse than methods without theoretical guarantees.

**Strengths:**

The proposed method is novel and interesting. Also, the geometric conditions are pretty clear. The conditions require that the manifold is well-sampled and sufficiently separated from other manifolds, with the sampling density given as a function of the curvature. The proofs in the paper are very clear and well written

The method is tested on the CIFAR datasets, showing competitive performance compared to state-of-the-art methods, even though it performs marginally worse than methods without theoretical guarantees.

**Weaknesses:**

Maybe will be useful give insights about how choose the hyperparameter $\lambda$.

**Questions:**

No questions.

**Limitations:**

Strategy of choosing $\lambda$

---

> ### Author Rebuttal · Authors · 2024-08-07
>
> Thank you for your strong support in the acceptance. Your comments are to the point, and we provide a reply below hoping to alleviate your concerns.
>
> **How to choose the hyperparameter $\lambda$ in practice:**
>
> Both reviewers XZ5T and Mh3Q pointed out this question. Our answer has two parts:
> - Guided by Lemma 3, if one has an estimate of the norm of the data points, curvature of the manifold, and within-manifold data density (i.e. $\zeta$), then the lower bound, $\lambda_l$, of $\lambda$ can be estimated. One can search $\lambda$s that are near to or greater than $\lambda_l$.
> - When the above quantities are difficult to know, one can alternatively rely on a heuristic grid search to find $\lambda$ on a training set, as we do in Appendix B.2.
>     - The use of $\lambda_0$ in line 672 of the paper follows from the work of [A] (their $\S$ 4.2.2) and [18] (their $\S$ 4.2 and Appendix F). The idea is that, $\lambda_0$ rules out a family of hyperparameters that lead to the solution $c$ being all zeros; such $c$’s are useless since they suggest that every point forms its own cluster.
>     - To figure out what $\alpha$ to use, we run experiments on CIFAR-100 data to grid search over $\alpha = \{2,5,10,20,50\}$. Since $\alpha = 20,50$ provided the best results, we used these values for the remaining experiments.
>
> We will include this discussion under Lemma 3, and refer to it in the experiments (section 3.2.2). Thank you for helping us make the paper more self-contained!
>
> Best Regards,
>
> Authors of Submission 17809
>
>
>
> [A] C. You, “Sparse Methods for Learning Multiple Subspaces from Large-scale, Corrupted and Imbalanced Data”, 2018.

---

> > ### Comment · Reviewer_XZ5T · 2024-08-13
> >
> > Thanks for your response!

---

### Author Rebuttal · Authors · 2024-08-07

Thanks to all the reviewers for their time and input on the paper. We appreciate that reviewers found the paper *novel and well-written* (XZ5T, Mh3Q), believe we provided *clear geometric conditions* (yHKx, XZ5T), *illustrated competitive performance* (XZ5T), and that our work is *relevant for many applications* (Mh3Q).

We address individual concerns below. Please feel free to communicate with us during the discussion phase if you have further concerns.

Thank you and regards,

Authors of Submission 17809

---

### Decision · Program_Chairs · 2024-09-25

**Decision:**

Accept (poster)

**Comment:**

The paper studies the problem of manifold clustering, in which we are given data on multiple (nonlinear) manifolds and would like to segment data based on their membership. The paper studies an approach to this problem inspired by subspace clustering, in which each data point is approximated by a sparse (approximately) affine combination of the other samples, and then spectral clustering is applied to the resulting coefficients. The paper’s main results are theoretical: it studies conditions under which solutions to this optimization problem are manifold-preserving, i.e., i.e., nonzero coefficients all correspond to samples from the correct manifold. The paper provides two different phrasings of these results: one in terms of convex bodies spanned by certain subsets of samples, and one in terms of manifold curvature. Experiments show that this method is applicable to learning datasets such as CIFAR-X, with better performance than subspace clustering methods (albeit a bit below state-of-the-art for these datasets).

Reviewers generally praised the theoretical novelty of the paper: although subspace clustering has been extensively analyzed, there is far less existing work on manifold clustering. Concerns focused on the choice of hyperparameter lambda, the role of the affine constraint in the formulation and the experiments. After considering author feedback, reviewers uniformly recommend acceptance: the paper provides a novel theoretical characterization of a natural computational approach to this important class of nonlinear models.

The AC concurs and recommends acceptance, with one caveat, as follows:

As a technical note, claims such as Lemma 2 (and hence Lemma 3) require not only assumptions on the *curvature* of the manifold, but on its *reach* [originally defined in a 1959 paper of Federer, and widely used in the theoretical analysis of algorithms for nonlinear data, e.g., manifold estimation / approximation]. Lemma 2 quantifies the error in a local, affine approximation to M_1. The claimed property only holds if the points from M_1 in the neighborhood B have small intrinsic distance to y. In general, M_1 can exit this neighborhood, ``loop around’’ and re-enter it, in which case the claimed bound can fail. If zeta(B) is smaller than the reach, this does not happen, points in the *extrinsic* neighborhood B are *intriniscally local*, in which case the conclusion of the lemma is correct. [A standard assumption would be that zeta(B) < reach(M_1); this is related to the paper’s stated assumption, since we always reach(M_1) <= 1/curvature(M_1), but ensures that the extrinsic neighborhood B is intrinsically local]

This issue should be carefully addressed in the final version.